a functional magnetic resonance imaging study. *R. Soc. open sci.* **6**: 181704.

Subject Areas:
neuroscience/behaviour/cognition

Keywords:
sleep, emotional regulation, reappraisal, fMRI

Author for correspondence:
Sandra Tamm
e-mail: sandra.tamm@ki.se

# Sleep restriction caused impaired emotional regulation without detectable brain activation changes—a functional magnetic resonance imaging study

Sandra Tamm[1,2], Gustav Nilsonne[1,2],
Johanna Schwarz[1,2], Armita Golkar[1,3],
Göran Kecklund[1,2], Predrag Petrovic[1], Håkan Fischer[3],
Torbjörn Åkerstedt[1,2] and Mats Lekander[1,2]

[1]Department of Clinical Neuroscience, Karolinska Institutet, Nobels väg 9, Stockholm 171 77, Sweden
[2]Stress Research Institute, and [3]Department of Psychology, Stockholm University, Stockholm, Sweden

 ST, 0000-0001-5371-9631; GN, 0000-0001-5273-0150

Sleep restriction has been proposed to cause impaired emotional processing and emotional regulation by inhibiting top-down control from prefrontal cortex to amygdala. Intentional emotional regulation after sleep restriction has, however, never been studied using brain imaging. We aimed here to investigate the effect of partial sleep restriction on emotional regulation through cognitive reappraisal. Forty-seven young (age 20–30) and 33 older (age 65–75) participants (38/23 with complete data and successful sleep intervention) performed a cognitive reappraisal task during fMRI after a night of normal sleep and after restricted sleep (3 h). Emotional downregulation was associated with significantly increased activity in the dorsolateral prefrontal cortex ($p_{FWE} < 0.05$) and lateral orbital cortex ($p_{FWE} < 0.05$) in young, but not in older subjects. Sleep restriction was associated with a decrease in self-reported regulation success to negative stimuli ($p < 0.01$) and a trend towards perceiving all stimuli as less negative ($p = 0.07$) in young participants. No effects of sleep restriction on brain

activity nor connectivity were found in either age group. In conclusion, our data do not support the idea of a prefrontal-amygdala disconnect after sleep restriction, and neural mechanisms underlying behavioural effects on emotional regulation after insufficient sleep require further investigation.

## 1. Introduction

Adequate sleep is important for emotional functioning, as indicated by a number of experimental studies (reviewed in [1]) and associations between sleep disturbance and mood disorders demonstrated in epidemiological studies [2–4]. Along these lines, increased emotional reactivity to negative emotional stimuli after experimental sleep deprivation has been shown in earlier studies [5–7]. Yoo *et al*. [6] proposed decreased connectivity between prefrontal control regions and amygdala as the underlying mechanism. Emotional responses can be regulated through a variety of strategies [8,9], including cognitively oriented strategies, such as cognitive reappraisal, that engage regions in the prefrontal cortex, proposed to inhibit activity in the amygdala [10]. However, whether sleep restriction affects the ability to explicitly regulate emotions through cognitive reappraisal [11] is not known. We here report a study where the effect of sleep restriction on cognitive reappraisal was tested in both younger and older subjects, motivated by observations that vulnerability to sleep deprivation, as well as emotional and cognitive functioning, change during an adult's lifetime [12–15].

One way to regulate an emotional response is to change the emotional meaning of the stimulus, i.e. to cognitively reappraise the stimulus [8]. Cognitive reappraisal has been studied repeatedly with functional magnetic resonance imaging (fMRI) [11,16–20]. Across studies (without sleep interventions), amygdala activity has been reduced when reappraising compared with passively viewing emotional stimuli. Prefrontal and parietal regions have been postulated as exerting top-down control during reappraisal, including posterior dorsomedial prefrontal cortex (dmPFC), dorsolateral prefrontal cortex (dlPFC), ventrolateral prefrontal cortex (vlPFC), lateral orbitofrontal cortex (lOFC) and posterior parietal cortex [11,16]. Some of the observed heterogeneity in activation patterns can putatively be explained by hidden moderators, e.g. heterogeneity in experimental paradigms, timing and instructions. Additionally, some heterogeneity is apparently due to differences in brain anatomical nomenclature. In a meta-analysis including only studies using stimuli from the International Affective Picture System (IAPS) [21], dlPFC and lOFC emerged as key areas [16], with the lOFC cluster partly overlapping with what has been reported as vlPFC in another later meta-analysis [11].

The potential importance of sleep for successful cognitive reappraisal has so far only been studied in terms of habitual sleep quality [22,23]. Minkel and colleagues found no relation between subjective sleep quality and BOLD responses nor self-reported success during cognitive reappraisal, but use of sleep medication was associated with less activity in mPFC and dlPFC during the task [22]. On the other hand, Mauss *et al*. suggested that poorer self-reported sleep quality was associated with a lower ability to decrease sadness using cognitive reappraisal [23]. The latter study, however, recorded only self-reports and not brain imaging measures. Moreover, these observational studies cannot rule out possible confounders such as psychiatric and somatic symptoms or psychosocial stress. In order to understand the causal effects of sleep on cognitive reappraisal, the use of experimental sleep manipulation is essential.

Another limitation of previous research on sleep and emotional processes is that almost exclusively younger individuals have been studied, despite findings that ageing alters emotional and cognitive functioning [12]. In addition, older individuals' sleep is shorter and less efficient (less sleep continuity, slow wave sleep and REM sleep) compared to younger [24], but they are, perhaps paradoxically, more resilient to sleep deprivation and show less cognitive impairment after sleep restriction, compared to younger [12]. In spite of some methodological challenges (i.e. difficulties for older adults in following instructions and other age-related confounders [25,26]), there is a need to involve older subjects in order to better represent the population of interest.

This study aimed primarily to investigate whether sleep restricted to 3 h (mimicking real-life partial sleep loss) affects emotional regulation through cognitive reappraisal in healthy adults on subjective ratings, brain activity measured with fMRI and psychophysiological outcomes. A secondary aim was to study the effects of age on emotional regulation. However, many older participants had difficulties following the specific instructions in the task. Therefore, this report focuses mainly on the younger participants, while results from the older subjects are reported for transparency. We specifically hypothesized that sleep restriction would lead to decreased self-rated success in emotional regulation

in response to negative stimuli, and that this effect would be associated with decreased activation of dlPFC and lOFC, increased amygdala activation and decreased connectivity between dlPFC/lOFC and amygdala[1].

# 2. Material and methods

Data for the present study were collected as part of the Stockholm Sleepy Brain project; a detailed description of design and procedures can be found in [27]. In brief, healthy participants underwent fMRI scanning on two occasions, about one month apart in a counterbalanced order, once after a full night's sleep and once after sleep restricted to 3 h. The experiment took place in the evening, starting between 17.00 and 20.00 and the full experiment lasted for about 3 h. Participants' sleep was monitored using polysomnography as well as subjective sleep measures. Researchers performing fMRI were blinded to participants' sleep condition.

## 2.1. Participants

Healthy participants were recruited through advertisements in newspapers and through the webpage www.studentkaninen.se. Fifty-three young and 44 older participants were invited to participate after an online screening procedure. Inclusion criteria were: no ferromagnetic objects in body, not claustrophobic, not pregnant, no refractive error exceeding 5 dioptres, not colour-blind, right-handed, to be 20–30 or 65–75 years old (inclusive), no current or past psychiatric or neurological illness, no hypertension nor diabetes, to not use psychoactive or immune-modulatory drugs, to not use nicotine every day and to drink four or fewer cups of coffee a day, fluency in Swedish and living in the greater Stockholm area. We excluded participants who had studied or had been occupied in the fields of psychology, behavioural science or medicine, including nursing and other related fields. The insomnia severity index (ISI) [28] and the depression subscale of the Hospital Anxiety and Depression scale (HADS) [29] and the Karolinska Sleep Questionnaire (KSQ) [30] were excluded due to pathological findings on MRI or after discovering fulfilling exclusion criteria after enrolment. Four young and four older participants were excluded due to pathological findings on MRI or discoveries fulfilling exclusion criteria after enrolment. One young and two older participants were unable to undergo the experiment because of feelings of claustrophobia, anxiety or panic. One young participant cancelled her participation due to a headache after the intervention night and one older participant cancelled his participation after the first scanning occasion. Forty-seven young and 37 older participants were scanned twice. For two older participants, the experiment had to be stopped due to technical reasons (at one respective session) and for five young participants, imaging data were lost for one session, due to a backup problem. Forty-two young and 35 older participants have complete data for both sessions.

For analyses regarding the effects of sleep, only participants with a successful intervention were included. Successful intervention was defined as more than 4 h sleep in the full sleep condition, less than 4 h in the sleep deprivation condition and a difference in total sleep time between the two conditions exceeding 2 h. Four young and four eligible older participants did not fulfil these criteria and were therefore not included in analyses of the effect of sleep restriction. See figure 1 for inclusion flowchart. The Karolinska Sleepiness Scale (KSS) [31] was used to assess sleepiness during the experiment.

## 2.2. Stimuli and fMRI paradigm

Forty-five negative and 15 neutral pictures were selected from IAPS [21]. Two trial lists, counterbalanced between sleep conditions, were used. Stimulus conditions were randomized in blocks of four, in order to balance conditions over the order of trials. The second trial list was constructed by reversing the first trial list, in an attempt to balance out any order effects. All the 15 neutral stimuli had the instruction 'maintain' in both trial lists, while the negative pictures had either 'maintain', 'upregulate' or 'downregulate' (15 of each), similarly to previous studies of reappraisal [17,32]. No negative picture had the same instruction in the two trial lists. The lists and scripts for presentation can be found at: https://doi.org/10.5281/zenodo.235595.

---

[1]A full list of hypotheses pre-conceived at registration of the Stockholm Sleep Brain Study can be found at: osf.io/zuf7t/.

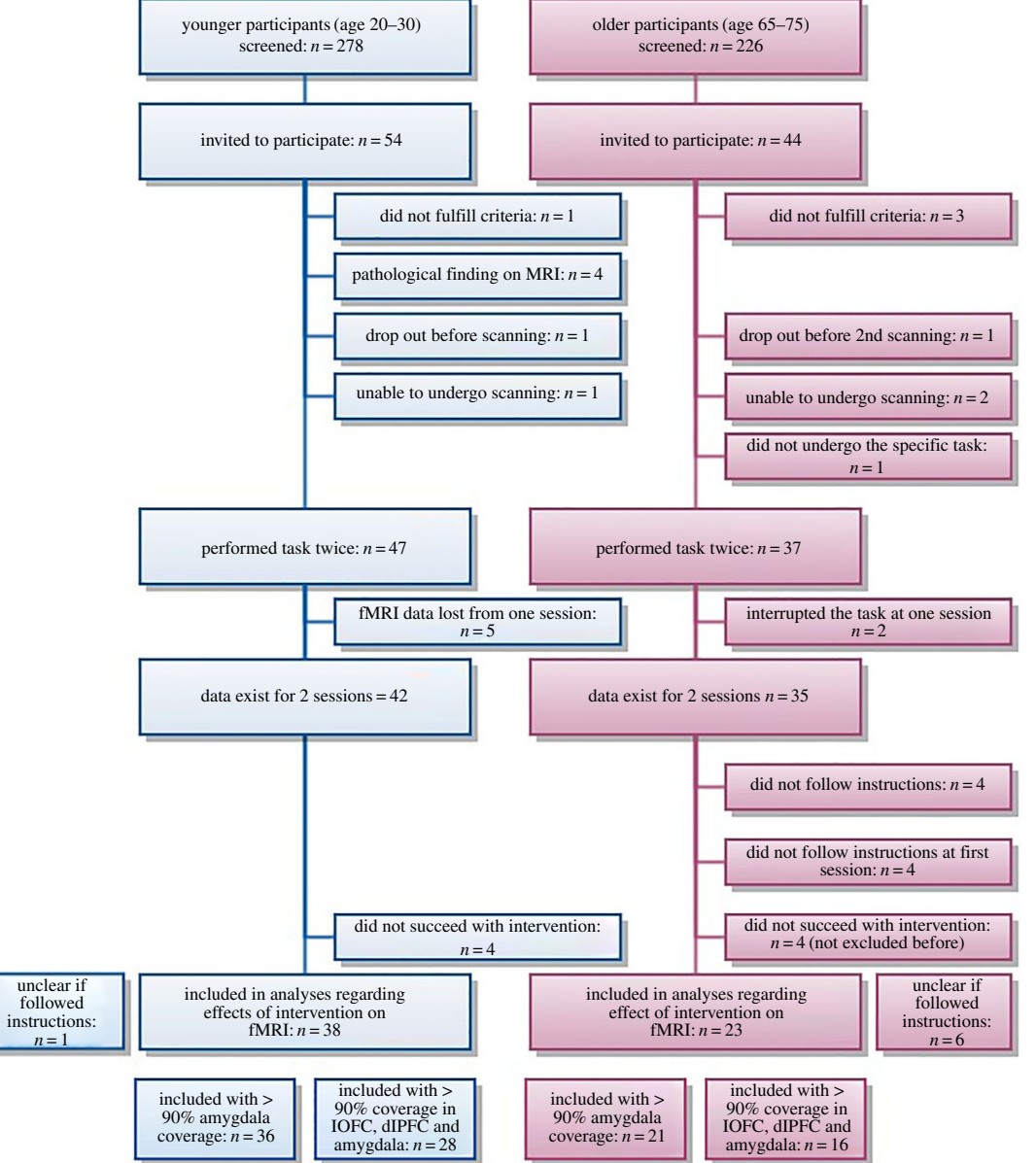

**Figure 1.** Inclusion flowchart.

In an instruction session before the experiment, participants were trained in how to perform the task using a separate set of stimuli. Following an arrow pointing upwards, participants were instructed to increase their emotional response to the following stimulus. After an arrow pointing downwards, they were instructed to decrease their emotional response. Lastly, following an arrow pointing to the right, they were instructed to just look and not change their spontaneous reaction (maintain). Participants were told to always look at the picture.

During fMRI, stimuli were shown using Presentation software (Neurobehavioral systems) displayed via fMRI-compatible goggles with an eye-tracker on the right eye (Arrington Research). Each session consisted of 60 trials (15 maintain neutral, 15 maintain negative, 15 upregulate negative and 15 downregulate negative) (figure 2). The stimuli were shown for 5 s following 2 s of instruction (arrow). After a stimulus was presented, a blank screen was shown for 2 s, whereafter the participants were asked to rate how well they succeeded with the task on a 7-point scale. A cursor was placed on 4, corresponding to average performance with 1 corresponding to the worst possible performance and 7 to the best. Heart rate was recorded using a pulse oximeter and pupil diameter was recorded using the eye-tracker.

After the experiment, all pictures were shown again to the participants outside the scanner. Participants were instructed to rate their perceived unpleasantness in response to each picture on a

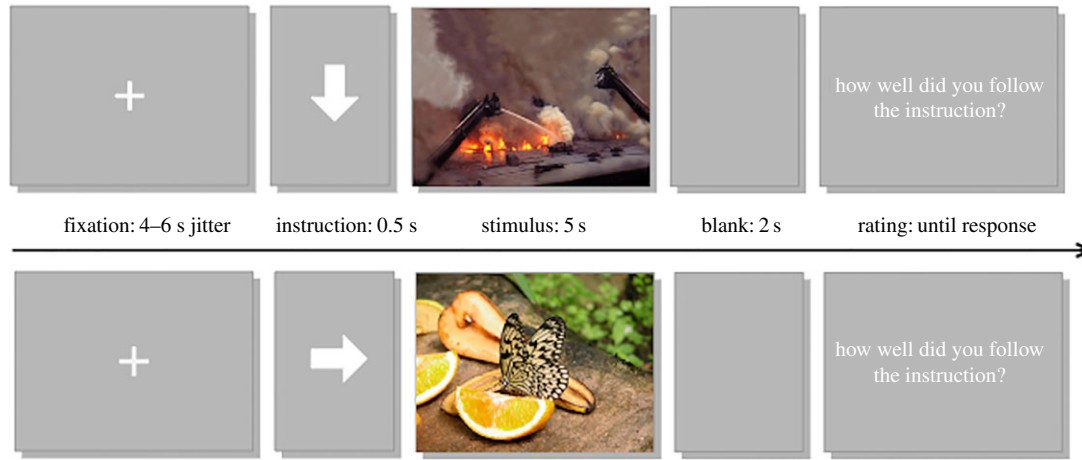

**Figure 2.** Experimental task. Stimuli were shown for 5 s following 2 s of instruction (arrow). After stimuli, a blank screen was shown for 2 s, and after that participants were asked to rate how well they succeeded with the task on a 7-point scale. A cursor was initially placed on '4'. Note: The stimuli shown in this figure are not IAPS pictures and were not included in the task.

7-point scale (1 = no unpleasantness, 7 = maximal unpleasantness). These ratings were added after the start of data collection, and therefore only 16 young and 35 old subjects have corresponding data for both sessions.

After the experiment, participants were interviewed regarding their strategies to reappraise and participants who apparently had not followed the instructions were excluded from the analyses. Most commonly, participants in such cases reported that they were rating unpleasantness instead of the success of regulation, forgot which arrow corresponded to which instruction, or could not explain the task instructions at all. Four older participants misunderstood the instructions at both sessions and were completely excluded from the analyses. Additionally, four participants misunderstood the instruction at their first session but followed them on their second session. Accordingly, session 1 was removed for these participants. For one young and seven older participants, it was not clear whether they followed the instruction or not. These participants were included in the analyses, but 'possibly did not understand instruction' (coded as 0 or 1) was included as a covariate in the analysis and tested for on the main contrasts of interest.

## 2.3. Final sample

Thirty-eight younger and 23 older participants could be included in intervention analyses with imaging data for the experimental task (figure 2). Where possible, additional subjects were included in analyses. For some of the whole-brain analyses, fewer participants were included because of poor brain coverage (see below).

## 2.4. Data acquisition

Imaging data were acquired using a 3.0 T scanner (Discovery MR750, GE), as described in detail elsewhere [27]. Functional scans were acquired in a gradient echo-planar-imaging (EPI) sequence, TR = 3 s, TE = 34 ms, flip angle = 80, 0.1 spacing and slice thickness 2.3. Field of view was placed so that the inferior border was at the lower margin of the pons. The sequence was optimized to cover the amygdala, but due to tilted heads in some subjects and human error, some subjects did not have full amygdala coverage, nor full coverage of the frontal cortex (see below).

## 2.5. Analysis of behavioural data

Behavioural data were analysed using R (http://www.R-project.org/). Scripts can be found at: https://doi.org/10.5281/zenodo.1434679. For mixed effects models, the main effects are reported as model estimates in original units (ratings from 1 to 7) with 95% CI. Significant interactions were followed by pair-wise comparisons (t-tests). Maintain neutral was considered reference, as well as full sleep and younger age in the models.

### 2.5.1. Rated success

The effect of stimulus type (maintain neutral, maintain/downregulate/upregulate negative) on rated success to follow the instruction after each stimulus presentation was investigated by mixed effects models stratified by age group. Stimulus type was modelled as a fixed effect and subject intercept as a random effect. Effects of sleep restriction on rated success were analysed stratified by age with stimulus type and sleep condition as interacting fixed effects and subject intercept as a random effect. To investigate age group effects on rated success in a model with all participants, the age group was added as a fixed effect, interacting with stimulus type.

### 2.5.2. Rated unpleasantness

As noted above, 35 older and 16 younger participants rated their perceived unpleasantness in response to all stimuli after the experiment and outside the scanner. Effects of valence (negative/neutral) were investigated stratified by age group. As for rated success, effects of sleep restriction were analysed stratified by age group. To investigate the effects of age group, the age group was added to the model including all participants.

### 2.5.3. Heart rate and pupil diameter

Heart rate and pupil diameter were analysed as measures of sympathetic activity. As in Nilsonne et al. [27], heart rate was determined based on recorded pulse events and was investigated within a time window of 4 s before each instruction to 10 s after pictures were shown. Time courses were inspected by two researchers independently (S.T., G.N.) for each participant, and recordings judged as excessively noisy were excluded ($n = 19$). Heart rates less than 40 beats per minute (bpm) or greater than 110 bpm were considered non-physiological and were censored. Heart rate was normalized to the heart rate 4 s before the arrow and averaged over the 5 s of stimulus and entered in a mixed effects model with stimulus type, age and sleep. Results are presented in the supplement.

As in Nilsonne et al. [27], to remove artefacts, all records of pupil height and width where the first derivative was less than $-3$ or greater than 3 were discarded, along with one consecutive data point before and after. Furthermore, all records of pupil height and width less than 0.1 cm and greater than 0.3 cm were discarded. If at least 50% of data remained in a window from 6 s before each event onset (4 s before arrow) to 10 s after, a loess curve was fitted to impute the missing data and down-sample the time-course for plotting. Pupil height and width were averaged over 5 s (during the stimuli) to yield a pupil diameter measure and this measure was entered into a mixed effects model. Results are presented in the electronic supplementary material.

## 2.6. fMRI preprocessing and analyses

Imaging data were analysed using SPM12 (Statistical parametric mapping, The Welcome Department of Imaging Neuroscience, Institute of Neurology, University College London) running on Matlab2015 (MATLAB 2015, The MathWorks, Inc., Massachusetts, USA). Preprocessing was done as described recently [33], including slice-time correction, realignment and unwarping, coregistration to the structural T1-weighted image and normalizing to MNI using a group-specific DARTEL template. Smoothing with a kernel with $8 \times 8 \times 8$ mm size at FWHM was performed. During quality check, it was discovered that five subjects had less than 90% coverage of amygdala in one session and 19 subjects had poor coverage of dlPFC and lOFC (see details below). For whole-brain analysis, we only included participants with greater than 90% coverage of each region (using ROIs described below).

Statistical analyses were performed using standard procedures for fMRI involving a fixed effects model at first level (one per session). This model included separate regressors for stimulus type as well as instruction type, which were convolved with the canonical hemodynamic response function. Rating events, as well as button presses and movement parameters from the realignment step, were included as regressors of no interest. The design matrix can be found in electronic supplementary material, figure S1.

At second level, one sample t-tests were performed to investigate the effects of stimulus type. Possible confounders were added to the t-tests and investigated through F contrasts. For whole-brain analyses, only subjects with greater than 90% coverage of regions of interest were included, resulting in lower numbers of participants ($n = 47$ younger and 34 older for negative > neutral and $n = 42$ younger and

30 older for regulate contrasts). To investigate effects of sleep restriction and age, a flexible factorial design was used ($n = 36$ young and 21 older for negative > neutral and $n = 28$ young and 16 older for regulate contrasts), but after a manipulation check (see below), we restricted the main analysis of the effect of sleep to the young participants. Thus, sleep restriction effects in young were investigated through paired $t$-tests.

A region of interest (ROI) analysis was performed to test the specific hypotheses regarding the effect of sleep restriction on amygdala, lOFC and dlPFC. For amygdala, we used an anatomical ROI based on the Automated Anatomical Labelling in the Wake Forest University (WFU) pickatlas toolbox in SPM. For dlPFC and lOFC, we used spherical ROIs based on peak coordinates from the meta-analysis from Kalisch [16], 15 mm for dlPFC and 10 mm lOFC (see electronic supplementary material, figure S2). Mean contrast values were extracted from these ROIs and entered into a mixed effects model. For this analysis, we also included participants with parts of the regions missing ($n = 38$ young and 23 older in total).

Psychophysiological interaction (PPI) analyses were performed to investigate the connectivity related to negative emotion and downregulating the emotional response. Time courses were extracted for seeds in bilateral amygdalae. We used peak coordinates for the contrast negative > neutral and a sphere of 6 mm radius around the peak. A PPI variable (the interaction term) was created for each amygdala and the contrasts negative > neutral and downregulate > maintain (4 in total) (see electronic supplementary material for design matrix). A second GLM analysis was performed with this PPI variable, the respective contrast and amygdala BOLD signal. The first-level contrasts were entered into one sample $t$-tests on second level to study the connectivity related to the task. An ROI analysis was performed for dlPFC and lOFC to investigate the effect of sleep restriction on amygdala connectivity to dlPFC and lOFC.

For completeness, all fMRI results are shown thresholded at $p = 0.001$ and with an extent threshold of 20 voxels. However, $p_{FWE} < 0.05$ was considered statistically significant, in line with conventions in the field. Anatomical areas were defined using the AAL in MRIcron. All statistical maps can be found on Neurovault (https://neurovault.org/collections/FWHMMCKI/) and all scripts at: https://doi.org/10.5281/zenodo.1434679.

# 3. Results

Demographic variables are shown in table 1. A more detailed report of the polysomnography results can be found in [34]. Because the drop-outs in this publication differ compared to previous publications from the same experiment, the numbers are slightly different compared to [27,33–35].

## 3.1. Task effects and manipulation check

Sleep restriction was associated with more sleepiness (higher KSS ratings), compared to the full sleep condition ($p < 0.001$, table 1), confirming the effect of the sleep manipulation.

### 3.1.1. Negative > neutral (maintain)

When contrasting negative to neutral pictures for the maintain instruction across age groups and sleep conditions, increased activity was found in clusters in the occipital gyri, a cluster in the precentral/frontal gyrus (right), in the middle/anterior cingulate cortex and in a cluster in the precentral gyrus extending in to insula (left) (figure 3$a$, table 2$a$). No significant effect in the amygdala was seen for negative > neutral. Results are presented separately for young and older in figure 3$b$,$c$ and table 3, with considerably smaller areas activated in older.

To better correspond to the effect of stimulus onset, a second model was investigated, where stimulus events were modelled with a duration of 0 s (stick-function). Negative > neutral stimuli with maintain instruction are presented in figure 3$d$ and table 2$b$. As expected, this model showed a similar result, but also revealed increased amygdala activity for negative compared to neutral stimuli, suggesting a more transient involvement of this structure.

### 3.1.2. Downregulate > maintain (negative)

Downregulate negative compared to maintain negative showed activation of prefrontal areas, including a cluster around the frontal gyrus extending into cingulate cortex and supplemental motor area, as well as bilateral clusters in orbitofrontal cortex/insula (figure 4$a$, table 4). These effects are displayed together

**Table 1.** Continuous values are reported as means with standard deviations, unless otherwise indicated. Categorical data are reported with percentages. Sleep measures are reported in minutes.

| variables | young | old |
|---|---|---|
| sample | | |
| number of subjects | 47 | 33 |
| demographics | | |
| age (median, interquartile range) | 23.0 (21.5 – 25.0) | 68.0 (67.0 – 71.0) |
| sex (females) | 24 (51.1%) | 17 (51.5%) |
| BMI | 22.9 ($\pm$ 3.1) | 24.6 ($\pm$ 3.0) |
| education | | |
| elementary school | 1 (2.1%) | 2 (6.1%) |
| high school | 10 (21.3%) | 14 (42.4%) |
| university degree | 6 (12.8%) | 16 (48.5%) |
| university student | 30 (63.8%) | 1 (3.0%) |
| HADS | | |
| depression | 1.1 ($\pm$ 1.4) | 1.2 ($\pm$ 1.0) |
| anxiety | 2.8 ($\pm$ 2.4) | 1.5 ($\pm$ 1.5) |
| sleep | | |
| insomnia severity index | 3.6 ($\pm$ 2.1) | 2.3 ($\pm$ 1.6) |
| Karolinska Sleepiness Scale, full sleep | 5.1 ($\pm$ 1.7) | 4.6 ($\pm$ 1.4) |
| Karolinska Sleepiness Scale, sleep restriction | 7.2 ($\pm$ 1.5) | 6.7 ($\pm$ 1.7) |
| total sleep time (min), full sleep | 429.1 ($\pm$ 77.4) | 396.1 ($\pm$ 64.8) |
| total sleep time (min), sleep restriction | 185.3 ($\pm$ 36.7) | 159.2 ($\pm$ 32.8) |
| REM sleep (min), full sleep | 86.8 ($\pm$ 29.9) | 76.6 ($\pm$ 38.0) |
| REM sleep (min), sleep restriction | 28.2 ($\pm$ 15.8) | 26.5 ($\pm$ 19.4) |
| slow wave sleep (min), full sleep | 98.0 ($\pm$ 32.0) | 39.1 ($\pm$ 32.6) |
| slow wave sleep, min (sleep restriction) | 70.5 ($\pm$ 16.5) | 29.6 ($\pm$ 26.1) |

with ROIs from the meta-analysis by Kalisch in figure 4b, showing high agreement. Young and older are presented separately in figure 4c,d and table 5. When contrasting maintain negative > downregulate negative, no effect in amygdala was seen (see electronic supplementary material, figure S3 and table S1 for complete results).

### 3.1.3. Upregulate > maintain (negative)

Upregulate compared to maintain was associated with increased activity in middle and anterior cingulate cortex, (table 6 and figure 5).

### 3.1.4. Covariates

Sex, test time type (whether participants were scanned earlier or later in the evening) or whether participants possibly misunderstood instructions did not notably affect the results for any of the main effects; statistical maps can be viewed at https://neurovault.org/collections/FWHMMCKI/.

### 3.1.5. Ratings

Participants rated how well they managed to follow the instruction after each stimulus (figure 6a,b). In young participants, higher success was rated for the maintain conditions compared to the regulate conditions; thus the highest success was reported for maintain neutral (mean 6.29) and decreasingly for maintain negative (−0.86 [−0.97, −0.75], $p < 0.001$), upregulate negative (−1.20 [−1.31, −1.09],

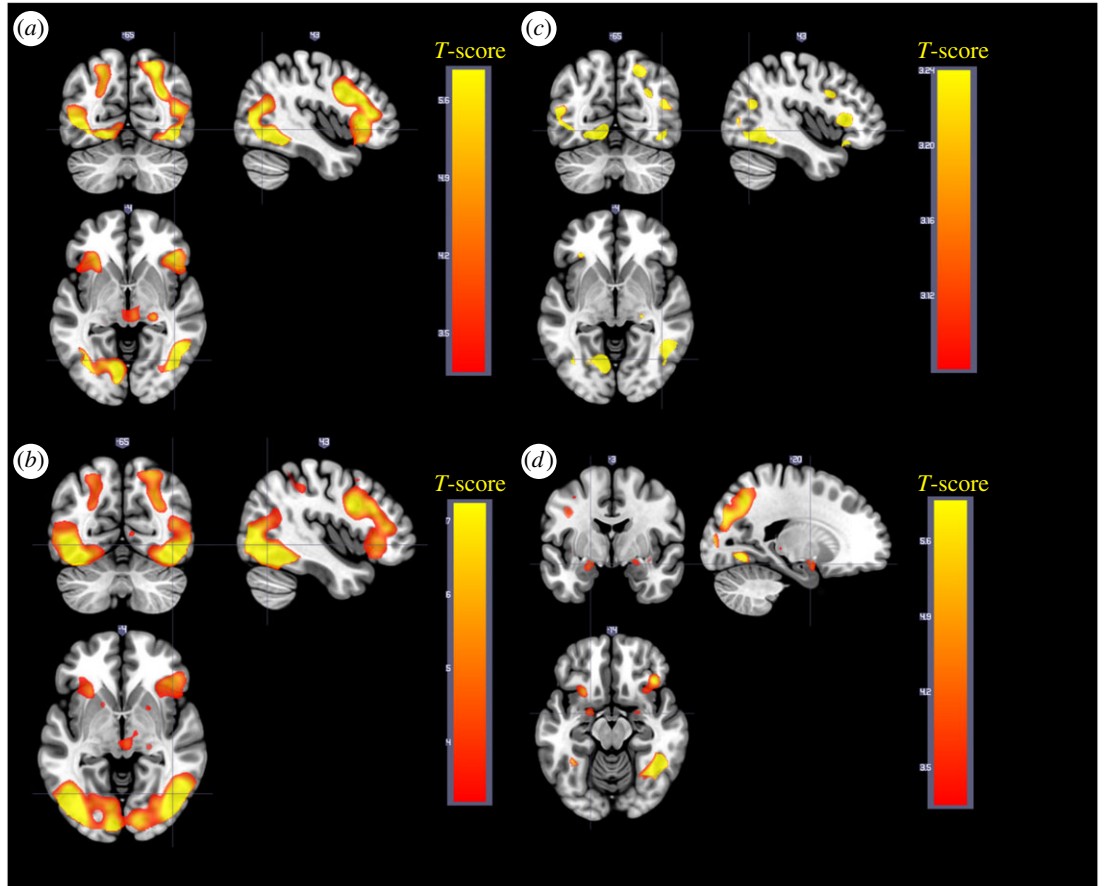

**Figure 3.** Main effect of the contrast negative > neutral for the maintain instruction. (*a*) Negative > neutral, all participants. (*b*) Negative > neutral, young participants. (*c*) Negative > neutral, older participants. (*d*) Negative > neutral, all participants. Stimuli modelled with a duration of 0 s.

$p < 0.001$) and downregulate negative (−1.63 [−1.74, −1.52], $p < 0.001$), compared to maintain neutral (numbers represent effect estimates from mixed effects models (ratings) and 95% CI). Older participants, across sleep conditions, rated highest success for maintain neutral (mean 6.12) and decreasingly for upregulate negative (−1.24 [−1.38, −1.10], $p < 0.001$), maintain negative (−2.41 [−2.54, −2.27], $p < 0.001$) and downregulate negative (22.74 [22.88, 22.60], $p < 0.001$) (figure 6*b*).

Ratings of unpleasantness are presented in figure 6*c,d*. Across sleep conditions, young participants reported less unpleasantness in response to the neutral pictures (mean 1.13) and higher unpleasantness in response to negative pictures (2.79 [2.66, 2.93], $p < 0.001$). Similarly, older participants reported less unpleasantness (mean 1.19) in response to neutral stimuli and higher unpleasantness in response to negative stimuli (4.06 [3.97, 4.16], $p < 0.001$).

As is shown in figure 4*c* and table 5*a*, young participants showed expected activity in dlPFC and lOFC when downregulating, confirming the validity of the paradigm. In older participants, the main effect of downregulating was not significant at $p_{FWE} < 0.05$ FWE in any cluster (figure 4*d* and table 5*b*). The ratings of success also indicated that young participants followed the instructions, as indicated by higher success for maintain compared to regulate, whereas this was not the case in older. Because of the higher proportion of older individuals misunderstanding the instructions, the indicated poor success in following the instructions, and non-significant activations in pre-registered regions of interest, the main analyses of the effect of sleep restriction on fMRI contrasts and ratings, were restricted to the young participants in the main text. For transparency, the age effect on ratings and fMRI was formally tested in full factorial designs, and the complete results are presented in electronic supplementary material, tables S2–S7 and figures S4–S9, as summarized below. Also, the effects of sleep restriction on fMRI across the whole sample, as well as the age × sleep interactions, are presented in electronic supplementary material, figures S10 and S11 and tables S8–S10.

**Table 2.** Negative > neutral, all, full duration.

| cluster | peak | peak | peak | peak | | | | MRIcron AAL (peak) | side |
| equivk | $p_{\text{(FWE-corr)}}$ | $T$ | equivZ | $p_{\text{(unc)}}$ | $x$ | $y$ | $z$ (mm) | | |
| --- | --- | --- | --- | --- | --- | --- | --- | --- | --- |
| (a) full stimulus duration | | | | | | | | | |
| 10 424 | <0.001 | 12.33 | Inf | <0.001 | 45 | −66 | −10 | inferior temporal | R |
| | <0.001 | 10.24 | Inf | <0.001 | 46 | −74 | 0 | middle occipital | R |
| | <0.001 | 8.55 | 7.68 | <0.001 | 28 | −74 | 34 | middle occipital | R |
| 8020 | <0.001 | 10.16 | Inf | <0.001 | −40 | −68 | −4 | inferior occipital | L |
| | <0.001 | 8.89 | Inf | <0.001 | −32 | −86 | 4 | middle occipital | L |
| | <0.001 | 8.8 | Inf | <0.001 | −36 | −76 | −2 | inferior occipital | L |
| 8791 | <0.001 | 9.32 | Inf | <0.001 | 45 | 8 | 30 | precentral | R |
| | <0.001 | 7.4 | 6.8 | <0.001 | 50 | 32 | 15 | inferior frontal, triangular part | R |
| | <0.001 | 7.1 | 6.56 | <0.001 | 48 | 26 | 24 | inferior frontal, triangular part | R |
| 6817 | <0.001 | 7.61 | 6.96 | <0.001 | 6 | 54 | 24 | frontal superior medial | R |
| | 0.009 | 5.12 | 4.9 | <0.001 | 9 | 26 | 33 | middle cingular | R |
| | 0.016 | 4.95 | 4.75 | <0.001 | −8 | 33 | 26 | anterior cingular | L |
| 6250 | 0 | 6.55 | 6.12 | <0.001 | −44 | 3 | 32 | precentral | L |
| | 0 | 6.11 | 5.75 | <0.001 | −30 | 26 | 2 | insula | L |
| | 0.006 | 5.2 | 4.97 | <0.001 | −42 | 32 | 18 | inferior frontal, triangular part | L |
| 160 | 0.001 | 5.79 | 5.48 | <0.001 | 22 | −26 | −2 | n.a. | |
| 582 | 0.004 | 5.34 | 5.09 | <0.001 | −63 | −30 | 36 | supramarginal | L |
| 782 | 0.007 | 5.15 | 4.93 | <0.001 | 14 | 14 | 8 | caudate | R |
| | 0.043 | 4.68 | 4.51 | <0.001 | 9 | −9 | 8 | thalamus | R |
| | 0.068 | 4.55 | 4.39 | <0.001 | 14 | 6 | 10 | n.a. | |
| 203 | 0.014 | 4.99 | 4.78 | <0.001 | 63 | −21 | 34 | supramarginal | R |

(Continued.)

| cluster equivk | peak $p_{(FWE-corr)}$ | peak $T$ | peak equivZ | peak $p(unc)$ | $x$ | $y$ | $z$ (mm) | MRIcron AAL (peak) | side |
|---|---|---|---|---|---|---|---|---|---|
| 925 | 0.025 | 4.83 | 4.64 | <0.001 | −2 | −48 | 27 | posterior cingular | L |
| | 0.524 | 3.83 | 3.73 | <0.001 | 6 | −54 | 45 | precuneus | R |
| 393 | 0.067 | 4.56 | 4.4 | <0.001 | 6 | −26 | −4 | n.a. | |
| | 0.439 | 3.91 | 3.81 | <0.001 | −4 | −20 | −8 | n.a. | |
| | 0.69 | 3.67 | 3.58 | <0.001 | 9 | −16 | −8 | n.a. | |
| 109 | 0.262 | 4.12 | 4 | <0.001 | −12 | 4 | 9 | n.a. | |
| 30 | 0.648 | 3.71 | 3.62 | <0.001 | 2 | −28 | −22 | n.a. | |
| 39 | 0.867 | 3.47 | 3.4 | <0.001 | −9 | −15 | 9 | thalamus | L |
| (b) stimulus modelled with 0 s duration | | | | | | | | | |
| 8311 | 0 | 13.49 | inf | 0 | −40 | −68 | −4 | inferior occipital | L |
| | 0 | 11.56 | inf | 0 | −48 | −68 | 2 | middle occipital | L |
| | 0 | 10.39 | inf | 0 | −40 | −76 | 6 | middle occipital | L |
| 9601 | 0 | 13.43 | inf | 0 | 45 | −62 | −9 | inferior temporal | R |
| | 0 | 10.4 | inf | 0 | 46 | −70 | −3 | inferior temporal | R |
| | 0 | 9.16 | inf | 0 | 27 | −74 | 34 | middle occipital | R |
| 6193 | 0 | 9.94 | inf | 0 | 46 | 8 | 30 | precentral | R |
| | 0 | 7.42 | 6.82 | 0 | 45 | 26 | 4 | inferior frontal, triangular part | R |
| | 0 | 6.34 | 5.94 | 0 | 39 | 28 | −14 | inferior frontal, orbital part | R |
| 1266 | 0 | 8.26 | 7.46 | 0 | −62 | −30 | 32 | supramarginal | L |
| 3656 | 0 | 7.38 | 6.78 | 0 | −45 | 3 | 30 | precentral | L |
| | 0 | 6.31 | 5.92 | 0 | −32 | 26 | 6 | insula | L |
| | 0.011 | 5.07 | 4.86 | 0 | −27 | 18 | −14 | insula | L |

(*Continued.*)

**Table 2.** (*Continued.*)

| cluster equivk | peak p(FWE-corr) | peak T | peak equivZ | peak p(unc) | x | y | z (mm) | MRIcron AAL (peak) | side |
|---|---|---|---|---|---|---|---|---|---|
| 410 | 0.001 | 5.74 | 5.44 | 0 | 63 | −21 | 32 | supramarginal | R |
| 1367 | 0.001 | 5.69 | 5.39 | 0 | 8 | 52 | 26 | superior medial frontal | R |
| | 0.035 | 4.76 | 4.58 | 0 | −9 | 51 | 20 | superior medial frontal | L |
| 991 | 0.002 | 5.52 | 5.25 | 0 | 4 | −28 | −4 | n.a. | |
| | 0.083 | 4.52 | 4.36 | 0 | 8 | −16 | −9 | n.a. | |
| | 0.554 | 3.83 | 3.73 | 0 | −10 | −22 | −9 | n.a. | |
| 295 | 0.002 | 5.5 | 5.23 | 0 | 14 | −78 | 6 | calcarine | R |
| 85 | 0.033 | 4.79 | 4.6 | 0 | 22 | −27 | −2 | n.a. | |
| 139 | 0.159 | 4.32 | 4.18 | 0 | −21 | −2 | −16 | amygdala | L |
| 144 | 0.159 | 4.32 | 4.18 | 0 | 9 | 24 | 33 | middle cingular | R |
| | 0.911 | 3.44 | 3.36 | 0 | 9 | 18 | 44 | middle cingular | R |
| 285 | 0.166 | 4.3 | 4.17 | 0 | −3 | −48 | 27 | posterior cingular | L |
| 73 | 0.287 | 4.11 | 3.99 | 0 | −2 | 6 | 33 | middle cingular | L |
| 56 | 0.395 | 3.99 | 3.88 | 0 | −9 | 16 | 44 | supplementary motor area | L |
| 64 | 0.606 | 3.78 | 3.69 | 0 | 34 | −8 | −10 | n.a. | |
| 57 | 0.647 | 3.74 | 3.65 | 0 | 14 | 12 | 9 | caudate | R |
| 40 | 0.737 | 3.66 | 3.57 | 0 | 22 | −3 | −12 | amygdala | R |

**Table 3.** Negative > neutral, young and older separately.

| cluster equivk | peak p(FWE-corr) | peak T | peak equivZ | peak p(unc) | x | y | z (mm) | MRIcron AAL (peak) | side |
|---|---|---|---|---|---|---|---|---|---|
| *(a) young* | | | | | | | | | |
| 32 401 | <0.001 | 12.7 | inf | <0.001 | 42 | −68 | −9 | inferior occipital | R |
| | <0.001 | 12.5 | inf | <0.001 | −42 | −72 | −8 | inferior occipital | L |
| | <0.001 | 11.04 | inf | <0.001 | 51 | −72 | −4 | inferior temporal | R |
| 8766 | <0.001 | 9.73 | inf | <0.001 | 46 | 9 | 30 | inferior frontal, opercular part | R |
| | <0.001 | 7.34 | 6.44 | <0.001 | 50 | 32 | 15 | inferior frontal, triangular part | R |
| | <0.001 | 7.32 | 6.43 | <0.001 | 48 | 24 | 27 | inferior frontal, triangular part | R |
| 7773 | <0.001 | 6.9 | 6.13 | <0.001 | 6 | 52 | 26 | superior medial frontal | R |
| | <0.001 | 6.59 | 5.91 | <0.001 | 4 | 42 | 39 | superior medial frontal | R |
| | <0.001 | 6.14 | 5.57 | <0.001 | −8 | 20 | 46 | supplementary motor area | L |
| 4257 | 0.001 | 6.09 | 5.53 | <0.001 | −42 | 3 | 32 | precentral | L |
| | 0.002 | 5.8 | 5.31 | <0.001 | −28 | 24 | 0 | insula | L |
| | 0.074 | 4.7 | 4.42 | <0.001 | −40 | 28 | 20 | inferior frontal, triangular part | L |
| 86 | 0.059 | 4.77 | 4.48 | <0.001 | 22 | −26 | −2 | n.a. | |
| 206 | 0.072 | 4.71 | 4.43 | <0.001 | 4 | −22 | −3 | n.a. | |
| | 0.822 | 3.65 | 3.51 | <0.001 | 12 | −14 | −6 | n.a. | |
| 194 | 0.106 | 4.59 | 4.32 | <0.001 | 10 | −10 | 9 | thalamus | R |
| 269 | 0.111 | 4.57 | 4.31 | <0.001 | −64 | −30 | 38 | supramarginal | L |
| 515 | 0.16 | 4.45 | 4.2 | <0.001 | 12 | 12 | 14 | caudate | R |
| | 0.714 | 3.77 | 3.61 | <0.001 | 20 | 8 | 0 | pallidum | R |
| | 0.82 | 3.65 | 3.51 | <0.001 | 16 | 4 | 12 | n.a. | R |
| 485 | 0.293 | 4.22 | 4.01 | <0.001 | 6 | −56 | 44 | precuneus | R |

(Continued.)

**Table 3.** (*Continued.*)

| cluster equivk | peak $p_{(FWE-corr)}$ | peak $T$ | peak equivZ | peak $p$(unc) | x | y | z (mm) | MRIcron AAL (peak) | side |
|---|---|---|---|---|---|---|---|---|---|
| | 0.361 | 4.14 | 3.94 | <0.001 | 4 | −57 | 33 | precuneus | R |
| 38 | 0.709 | 3.78 | 3.62 | <0.001 | −45 | 22 | 42 | middle frontal | L |
| 82 | 0.748 | 3.73 | 3.58 | <0.001 | 26 | −3 | 46 | n.a. | |
| | 0.938 | 3.46 | 3.34 | <0.001 | 33 | 2 | 46 | n.a. | |
| 46 | 0.793 | 3.68 | 3.54 | <0.001 | 2 | −27 | −22 | n.a. | |
| 30 | 0.899 | 3.54 | 3.41 | <0.001 | 40 | 12 | 54 | middle frontal | R |
| *(b)* old | | | | | | | | | |
| 1184 | 0.001 | 6.3 | 5.48 | <0.001 | −12 | −69 | −6 | lingual | L |
| | 0.929 | 3.58 | 3.38 | <0.001 | −10 | −90 | 2 | calcarine | L |
| 1220 | 0.001 | 6.11 | 5.35 | <0.001 | 50 | −52 | −3 | inferior temporal | R |
| | 0.002 | 6.06 | 5.31 | <0.001 | 45 | −57 | −10 | inferior temporal | R |
| | 0.031 | 5.17 | 4.67 | <0.001 | 46 | −44 | −14 | inferior temporal | R |
| 435 | 0.017 | 5.36 | 4.81 | <0.001 | 44 | 24 | 6 | inferior frontal, triangular part | R |
| 936 | 0.047 | 5.03 | 4.57 | <0.001 | −48 | −63 | −2 | middle temporal | L |
| | 0.063 | 4.94 | 4.49 | <0.001 | −40 | −68 | −4 | inferior occipital | L |
| | 0.109 | 4.74 | 4.34 | <0.001 | −46 | −74 | 10 | middle occipital | L |
| 467 | 0.092 | 4.8 | 4.39 | <0.001 | 24 | −69 | 51 | superior parietal | R |
| 219 | 0.108 | 4.75 | 4.35 | <0.001 | −44 | 21 | −12 | inferior frontal, orbital part | L |
| 79 | 0.325 | 4.32 | 4 | <0.001 | 44 | 24 | −16 | inferior frontal, orbital part | R |
| 203 | 0.358 | 4.27 | 3.97 | <0.001 | 44 | −62 | 20 | middle temporal | R |
| 158 | 0.399 | 4.22 | 3.93 | <0.001 | 62 | −22 | 33 | supramarginal | R |
| | 0.633 | 3.97 | 3.72 | <0.001 | 63 | −16 | 27 | supramarginal | R |

(*Continued.*)

**Table 3.** (*Continued.*)

| cluster equivk | peak $p_{(FWE-corr)}$ | peak T | peak equivZ | peak $p(unc)$ | x | y | z (mm) | MRIcron AAL (peak) | side |
|---|---|---|---|---|---|---|---|---|---|
| 179 | 0.44 | 4.18 | 3.89 | <0.001 | 28 | −68 | 32 | middle occipital | R |
| 167 | 0.441 | 4.17 | 3.89 | <0.001 | −3 | −48 | 28 | posterior cingular | L |
| 250 | 0.569 | 4.03 | 3.77 | <0.001 | −60 | −36 | 30 | supramarginal | L |
| | 0.858 | 3.7 | 3.49 | <0.001 | −60 | −28 | 34 | supramarginal | L |
| 153 | 0.63 | 3.97 | 3.72 | <0.001 | 39 | 9 | 32 | inferior frontal, opercular part | R |
| 213 | 0.689 | 3.91 | 3.67 | <0.001 | −45 | 12 | 27 | inferior frontal, opercular part | L |
| 45 | 0.716 | 3.88 | 3.64 | <0.001 | −6 | −21 | −9 | n.a. | |
| 179 | 0.745 | 3.85 | 3.62 | <0.001 | 8 | 54 | 26 | superior medial frontal | R |
| 36 | 0.777 | 3.81 | 3.58 | <0.001 | −32 | 20 | −26 | superior temporal pole | L |
| 111 | 0.823 | 3.75 | 3.53 | <0.001 | 8 | 27 | 32 | middle cingular | R |
| 24 | 0.909 | 3.62 | 3.42 | <0.001 | 12 | 8 | 4 | n.a. | |
| 57 | 0.916 | 3.6 | 3.41 | <0.001 | −32 | 32 | −6 | inferior frontal, orbital part | L |
| | 0.927 | 3.58 | 3.39 | <0.001 | −30 | 28 | 3 | insula | L |
| 22 | 0.957 | 3.5 | 3.32 | <0.001 | 54 | 20 | 3 | inferior frontal, opercular part | R |

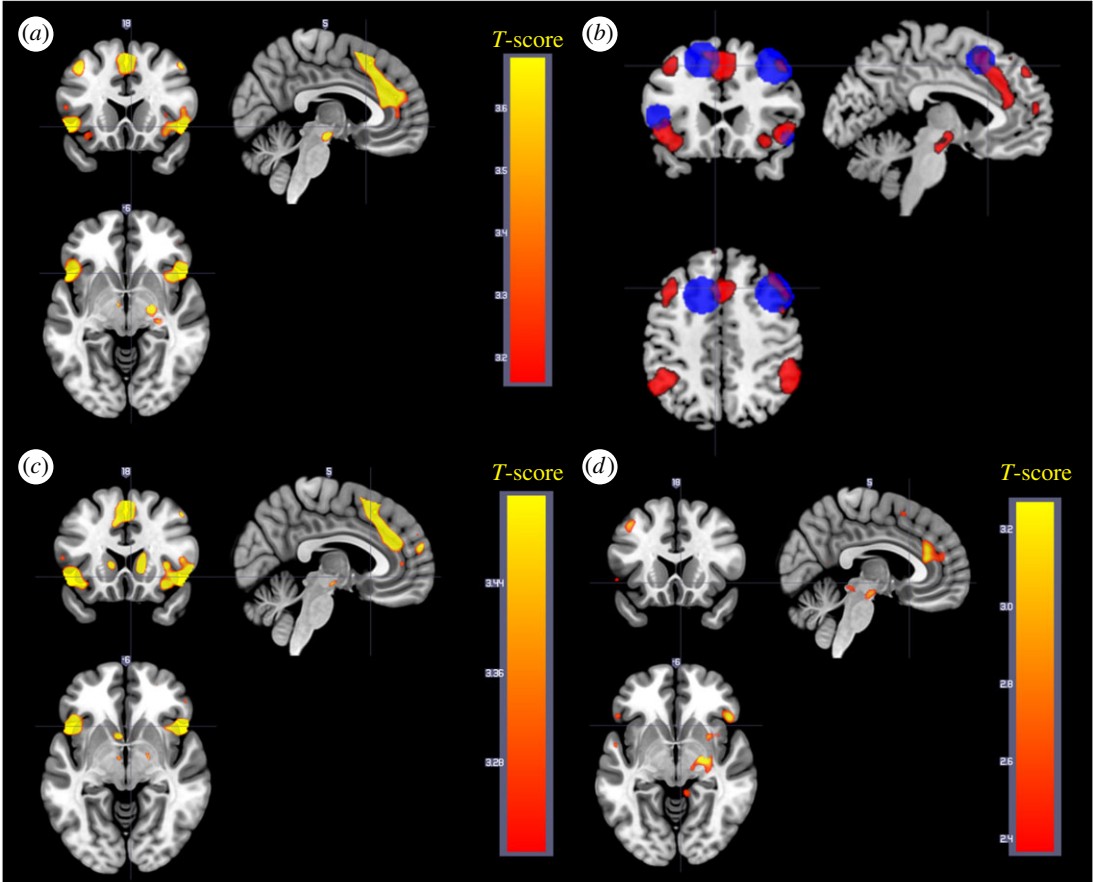

**Figure 4.** Downregulate > maintain for negative pictures (*a*) All participants. (*b*) Our data (in red), regions of interest from Kalish meta-analysis indicated in blue. (*c*) Downregulate > maintain, young participants. (*d*) Downregulate > maintain, older participants.

## 3.2. Effects of sleep restriction on rated success and rated unpleasantness (in young)

After sleep restriction, young participants reported decreased success in following the instructions, demonstrated as a significant main effect across stimulus types ($-0.27$ [$-0.43$, $-0.10$], $p = 0.002$, figure 6*a*). There was also a significant sleep condition $\times$ stimulus type interaction ($p = 0.039$). When decomposed, sleep restriction caused no significant effect on maintain negative ($-0.20$ [$-0.53$, $0.13$], $t_{42} = -1.2$, $p = 0.23$), but significantly decreased ratings of success for downregulate negative ($-0.50$ [$-0.77$, $-0.23$], $t_{42} = -3.75$, $p < 0.001$), maintain neutral ($-0.27$ [$-0.46$, $-0.07$], $t_{42} = -2.72$, $p = 0.01$) and upregulate negative ($-0.43$ [$-0.71$, $-0.15$], $t_{42} = -3.14$, $p < 0.01$).

In young participants, sleep restriction caused a borderline significant decrease in rated unpleasantness in response to all stimuli (20.17 [20.45, 0.10], $p > 0.07$, figure 6*c*). The valence $\times$ sleep condition interaction was not significant ($p = 0.75$).

## 3.3. Effect of sleep restriction on BOLD responses (in young)

Sleep restriction did not have any significant effect on the contrast maintain negative > maintain neutral when performing whole-brain analyses. An ROI analysis was performed on the average contrast value for amygdala bilaterally. The effect of sleep restriction on amygdala activity for the contrast maintain negative > maintain neutral was not significant; left ($-0.07$ [$-0.27$, $0.13$], $p = 0.47$), right ($-0.06$ [$-0.23$, $0.12$], $p = 0.47$). We investigated the effect of sleep restriction on the alternative model where stimuli were modelled with a duration of 0 s with a similar, non-significant, result.

### 3.3.1. Effect of sleep restriction on fMRI contrast downregulate > maintain negative (in young)

Sleep restriction did not have any significant effect on the contrast downregulate > maintain (negative) when performing whole-brain analyses. ROI analyses showed no significant effect of sleep restriction on

**Table 4.** Downregulate > maintain (negative), all.

| cluster equivk | peak p(FWE-corr) | peak T | peak equivZ | peak p(unc) | x | y | z (mm) | | |
|---|---|---|---|---|---|---|---|---|---|
| 3840 | 0.002 | 5.57 | 5.25 | <0.001 | 2 | 38 | 22 | anterior cingular | R |
| | 0.008 | 5.23 | 4.96 | <0.001 | 0 | 32 | 33 | superior medial frontal | L |
| | 0.017 | 5.02 | 4.78 | <0.001 | 2 | 16 | 56 | supplementary motor area | R |
| 1357 | 0.004 | 5.44 | 5.14 | <0.001 | 48 | 22 | −8 | inferior frontal, orbital part | R |
| | 0.334 | 4.11 | 3.97 | <0.001 | 51 | 12 | 8 | inferior frontal, opercular part | R |
| | 0.607 | 3.84 | 3.72 | <0.001 | 33 | 22 | −14 | insula | R |
| 2673 | 0.007 | 5.27 | 5 | <0.001 | −39 | −48 | 39 | inferior parietal | L |
| | 0.017 | 5.04 | 4.79 | <0.001 | −52 | −60 | 36 | angular | L |
| | 0.186 | 4.33 | 4.17 | <0.001 | −52 | −54 | 48 | inferior parietal | L |
| 2559 | 0.021 | 4.98 | 4.74 | <0.001 | 52 | −51 | 42 | inferior parietal | R |
| | 0.075 | 4.61 | 4.42 | <0.001 | 46 | −42 | 39 | inferior parietal | R |
| | 0.351 | 4.09 | 3.96 | <0.001 | 45 | −44 | 30 | angular | R |
| 1121 | 0.031 | 4.87 | 4.65 | <0.001 | −50 | 16 | −3 | n.a. | |
| | 0.797 | 3.65 | 3.55 | <0.001 | −36 | 21 | −15 | inferior frontal, orbital part | L |
| 396 | 0.035 | 4.83 | 4.62 | <0.001 | 21 | −12 | −8 | n.a. | |
| | 0.291 | 4.17 | 4.02 | <0.001 | 28 | −24 | −3 | n.a. | |
| | 0.861 | 3.57 | 3.48 | <0.001 | 20 | −21 | −9 | n.a. | |
| 1505 | 0.055 | 4.71 | 4.5 | <0.001 | 38 | 32 | 40 | middle frontal | R |
| | 0.071 | 4.63 | 4.43 | <0.001 | 38 | 48 | 24 | middle frontal | R |
| | 0.112 | 4.49 | 4.31 | <0.001 | 39 | 28 | 33 | middle frontal | R |
| 406 | 0.082 | 4.59 | 4.4 | <0.001 | −6 | −15 | −15 | n.a. | |
| | 0.116 | 4.48 | 4.3 | <0.001 | −6 | −9 | −9 | n.a. | |

(Continued.)

**Table 4.** (*Continued.*)

| cluster equiv $k$ | peak $p_{\text{(FWE-corr)}}$ | peak $T$ | peak equivZ | peak $p$(unc) | $x$ | $y$ | $z$ (mm) | | |
|---|---|---|---|---|---|---|---|---|---|
| | 0.214 | 4.28 | 4.12 | <0.001 | 6 | −15 | −14 | n.a. | |
| 456 | 0.104 | 4.52 | 4.33 | <0.001 | −39 | 21 | 48 | middle frontal | L |
| | 0.827 | 3.61 | 3.52 | <0.001 | −44 | 9 | 50 | precentral | L |
| 344 | 0.392 | 4.05 | 3.92 | <0.001 | 16 | 51 | 36 | superior frontal | R |
| | 0.945 | 3.42 | 3.34 | <0.001 | 18 | 57 | 28 | superior frontal | R |
| 1146 | 0.456 | 3.98 | 3.85 | <0.001 | −33 | 51 | 16 | middle frontal | L |
| | 0.509 | 3.93 | 3.81 | <0.001 | −30 | 44 | 14 | middle frontal | L |
| | 0.556 | 3.89 | 3.77 | <0.001 | −18 | 48 | 33 | superior frontal | L |
| 22 | 0.697 | 3.75 | 3.64 | <0.001 | −10 | 38 | 51 | superior medial frontal | L |
| 86 | 0.706 | 3.74 | 3.64 | <0.001 | −8 | 51 | 40 | superior medial frontal | L |
| 28 | 0.869 | 3.56 | 3.46 | <0.001 | −16 | −22 | −15 | parahippocampal | L |
| 39 | 0.904 | 3.51 | 3.42 | <0.001 | 45 | 46 | −3 | inferior frontal, orbital part | R |
| 39 | 0.907 | 3.5 | 3.41 | <0.001 | 15 | 12 | 12 | caudate | R |
| 29 | 0.94 | 3.43 | 3.35 | <0.001 | −51 | 16 | 10 | inferior frontal, opercular part | L |

**Table 5.** Downregulate > maintain, young and older.

| cluster equivk | peak $p_{(FWE-corr)}$ | peak T | peak equivZ | peak $p(unc)$ | x | y | z (mm) | MRIcron AAL (peak) | side |
|---|---|---|---|---|---|---|---|---|---|
| *(a)* young | | | | | | | | | |
| 3050 | 0.008 | 5.47 | 4.99 | <0.001 | −3 | 32 | 34 | middle cingular | L |
| | 0.012 | 5.36 | 4.91 | <0.001 | 0 | 39 | 21 | anterior cingular | L |
| | 0.079 | 4.79 | 4.45 | <0.001 | 2 | 18 | 54 | supplementary motor area | R |
| 1931 | 0.016 | 5.27 | 4.84 | <0.001 | 54 | −52 | 46 | inferior parietal | R |
| 2018 | 0.05 | 4.94 | 4.57 | <0.001 | 40 | 28 | 30 | inferior frontal, triangular part | R |
| | 0.117 | 4.66 | 4.35 | <0.001 | 38 | 30 | 42 | middle frontal | R |
| | 0.193 | 4.48 | 4.2 | <0.001 | 36 | 50 | 24 | middle frontal | R |
| 1072 | 0.066 | 4.85 | 4.5 | <0.001 | −44 | 21 | −9 | inferior frontal, orbital part | L |
| 1386 | 0.102 | 4.7 | 4.38 | <0.001 | 48 | 20 | −8 | inferior frontal, orbital part | R |
| | 0.299 | 4.31 | 4.06 | <0.001 | 51 | 14 | 4 | inferior frontal, opercualr part | R |
| | 0.612 | 3.97 | 3.77 | <0.001 | 34 | 20 | −14 | insula | R |
| 680 | 0.157 | 4.55 | 4.26 | <0.001 | 12 | 16 | 6 | caudate | R |
| | 0.786 | 3.79 | 3.61 | <0.001 | 15 | 3 | 16 | caudate | R |
| | 0.844 | 3.71 | 3.55 | <0.001 | 15 | 0 | 8 | n.a. | |
| 1356 | 0.204 | 4.46 | 4.18 | <0.001 | −52 | −60 | 36 | angular | L |
| | 0.29 | 4.33 | 4.07 | <0.001 | −39 | −48 | 39 | inferior parietal | L |
| | 0.51 | 4.07 | 3.85 | <0.001 | −52 | −58 | 48 | inferior parietal | L |
| 494 | 0.236 | 4.41 | 4.14 | <0.001 | −12 | 14 | 4 | caudate | L |
| | 0.528 | 4.05 | 3.84 | <0.001 | −6 | 10 | −6 | caudate | L |
| | 0.88 | 3.66 | 3.5 | <0.001 | −2 | −3 | 9 | n.a. | |
| 672 | 0.475 | 4.11 | 3.88 | <0.001 | −9 | 51 | 42 | superior medial frontal | L |

(*Continued.*)

**Table 5.** (*Continued.*)

| cluster equivk | peak $p_{\text{(FWE-corr)}}$ | peak T | peak equivZ | peak p(unc) | x | y | z (mm) | MRIcron AAL (peak) | side |
|---|---|---|---|---|---|---|---|---|---|
| | 0.57 | 4.01 | 3.8 | <0.001 | −34 | 54 | 20 | middle frontal | L |
| | 0.582 | 4 | 3.79 | <0.001 | −21 | 50 | 32 | middle frontal | L |
| 162 | 0.693 | 3.89 | 3.7 | <0.001 | −4 | −10 | −10 | n.a. | |
| | 0.974 | 3.44 | 3.3 | <0.001 | 6 | −12 | −10 | n.a. | |
| 246 | 0.721 | 3.86 | 3.67 | <0.001 | 6 | 60 | 20 | superior medial frontal | R |
| | 0.978 | 3.42 | 3.29 | 0.001 | 0 | 60 | 28 | superior medial frontal | L |
| | 0.98 | 3.41 | 3.28 | 0.001 | −2 | 58 | 15 | superior medial frontal | L |
| 124 | 0.754 | 3.82 | 3.64 | <0.001 | −30 | 0 | 58 | precentral | L |
| 40 | 0.819 | 3.75 | 3.57 | <0.001 | −10 | 38 | 51 | superior medial frontal | L |
| 67 | 0.898 | 3.63 | 3.47 | <0.001 | 9 | 40 | 3 | anterior cingular | R |
| 44 | 0.917 | 3.6 | 3.44 | <0.001 | −8 | −68 | 54 | precuneus | L |
| 36 | 0.954 | 3.51 | 3.37 | <0.001 | 12 | −69 | 46 | precuneus | R |
| 40 | 0.964 | 3.48 | 3.34 | <0.001 | −36 | 45 | 2 | middle frontal | L |
| 26 | 0.964 | 3.48 | 3.34 | <0.001 | 51 | 39 | −4 | inferior frontal, orbital part | R |
| 30 | 0.989 | 3.35 | 3.22 | 0.001 | −44 | 10 | 46 | precentral | L |
| (b) old | | | | | | | | | |
| 68 | 0.619 | 4.14 | 3.81 | <0.001 | 22 | −12 | −6 | n.a. | |
| 50 | 0.805 | 3.93 | 3.64 | <0.001 | 46 | 26 | −8 | inferior frontal, orbital part | R |
| 55 | 0.932 | 3.72 | 3.47 | <0.001 | −50 | −28 | 51 | postcentral | L |
| 29 | 0.979 | 3.55 | 3.33 | <0.001 | 8 | 34 | 24 | anterior cingular | R |

**Table 6.** Upregulate > maintain, all, young, older.

| cluster equivk | cluster p(unc) | peak p(FWE-corr) | peak T | peak equivZ | peak p(unc) | x | y | z (mm) | MRIcron AAL (peak) | side |
|---|---|---|---|---|---|---|---|---|---|---|
| (a) All | | | | | | | | | | |
| 310 | 0.064 | 0.235 | 4.19 | 4.05 | <0.001 | −2 | 22 | 38 | middle cingular | L |
| | | 0.856 | 3.51 | 3.42 | <0.001 | 2 | 32 | 28 | anterior cingular | R |
| 375 | 0.044 | 0.328 | 4.07 | 3.93 | <0.001 | 2 | −24 | 18 | n.a. | |
| | | 0.439 | 3.94 | 3.82 | <0.001 | 0 | −38 | 2 | n.a. | |
| 298 | 0.069 | 0.412 | 3.97 | 3.85 | <0.001 | 50 | 14 | 2 | inferior frontal, opercular part | R |
| 161 | 0.17 | 0.431 | 3.95 | 3.83 | <0.001 | 2 | 15 | 57 | supplementary motor area | R |
| 105 | 0.263 | 0.597 | 3.79 | 3.68 | <0.001 | 9 | −12 | −15 | n.a. | |
| 231 | 0.105 | 0.718 | 3.67 | 3.57 | <0.001 | −46 | 12 | −4 | insula | L |
| 38 | 0.508 | 0.773 | 3.61 | 3.52 | <0.001 | −8 | −15 | −16 | n.a. | |
| (b) Young | | | | | | | | | | |
| 622 | 0.01 | 0.018 | 5.2 | 4.78 | <0.001 | −3 | −33 | 2 | n.a. | |
| 121 | 0.21 | 0.106 | 4.64 | 4.33 | <0.001 | 2 | −24 | 20 | n.a. | |
| 373 | 0.036 | 0.145 | 4.53 | 4.24 | <0.001 | −21 | −45 | 21 | n.a. | |
| | | 0.883 | 3.59 | 3.44 | <0.001 | −27 | −50 | 15 | n.a. | |
| | | 0.937 | 3.49 | 3.35 | <0.001 | −30 | −38 | 6 | n.a. | |
| 279 | 0.065 | 0.155 | 4.51 | 4.22 | <0.001 | 2 | 10 | 62 | supplementary motor area | R |
| 130 | 0.194 | 0.158 | 4.5 | 4.22 | <0.001 | 20 | −14 | 28 | n.a. | |
| 342 | 0.044 | 0.207 | 4.4 | 4.13 | <0.001 | −2 | 22 | 39 | middle cingular | L |
| | | 0.91 | 3.55 | 3.4 | <0.001 | 2 | 32 | 28 | anterior cingular | R |
| 210 | 0.105 | 0.589 | 3.93 | 3.74 | <0.001 | −42 | 16 | −9 | insula | L |
| 104 | 0.243 | 0.671 | 3.85 | 3.66 | <0.001 | −20 | −28 | 27 | n.a. | L |

(Continued.)

**Table 6.** (*Continued.*)

| cluster equivk | cluster p(unc) | peak p(FWE-corr) | peak T | peak equivZ | peak p(unc) | x | y | z (mm) | MRIcron AAL (peak) | side |
|---|---|---|---|---|---|---|---|---|---|---|
| 56 | 0.393 | 0.714 | 3.8 | 3.62 | <0.001 | 21 | −28 | 28 | n.a. | |
| 34 | 0.511 | 0.844 | 3.65 | 3.49 | <0.001 | −15 | −9 | 28 | n.a. | |
| 30 | 0.54 | 0.864 | 3.62 | 3.47 | <0.001 | 36 | −38 | 2 | n.a. | |
| 24 | 0.588 | 0.919 | 3.53 | 3.38 | <0.001 | −12 | −27 | −15 | n.a. | |
| 36 | 0.498 | 0.927 | 3.51 | 3.37 | <0.001 | −32 | 50 | 24 | middle frontal | L |
| 51 | 0.415 | 0.932 | 3.5 | 3.36 | <0.001 | 50 | 14 | 2 | inferior frontal, opercular part | R |
| 12 | 0.715 | 0.943 | 3.48 | 3.34 | <0.001 | −38 | 12 | 14 | inferior frontal, opercular part | L |
| 18 | 0.645 | 0.966 | 3.41 | 3.27 | 0.001 | 38 | 18 | 12 | inferior frontal, opercular part | R |
| 9 | 0.758 | 0.981 | 3.34 | 3.21 | 0.001 | 18 | −40 | 21 | n.a. | |
| 6 | 0.81 | 0.984 | 3.32 | 3.2 | 0.001 | 14 | −32 | −12 | cerebellum | R |
| 8 | 0.774 | 0.988 | 3.29 | 3.17 | 0.001 | 26 | −44 | 15 | n.a. | |
| 3 | 0.876 | 0.99 | 3.27 | 3.15 | 0.001 | −21 | 2 | 32 | n.a. | |
| 3 | 0.876 | 0.99 | 3.27 | 3.15 | 0.001 | 32 | −48 | 10 | n.a. | |
| 6 | 0.81 | 0.991 | 3.26 | 3.14 | 0.001 | 0 | −8 | −6 | n.a. | |
| 1 | 0.938 | 0.991 | 3.26 | 3.14 | 0.001 | 10 | −18 | 27 | n.a. | |
| 1 | 0.938 | 0.994 | 3.23 | 3.11 | 0.001 | 44 | 14 | 8 | inferior frontal, opercular part | R |
| (c) Older | | | | | | | | | | |
| 25 | 0.565 | 0.864 | 3.76 | 3.5 | <0.001 | 9 | −15 | −15 | n.a. | |
| 23 | 0.583 | 0.882 | 3.73 | 3.48 | <0.001 | −10 | −18 | −14 | n.a. | |
| 6 | 0.802 | 0.982 | 3.45 | 3.24 | 0.001 | −14 | 21 | 56 | superior frontal | L |
| 7 | 0.783 | 0.986 | 3.42 | 3.22 | 0.001 | 24 | −27 | 68 | precentral | R |
| 22 | 0.592 | 0.987 | 3.41 | 3.21 | 0.001 | 42 | 28 | −8 | inferior frontal, orbital part | R |
| 2 | 0.899 | 0.993 | 3.36 | 3.16 | 0.001 | 33 | 51 | 14 | middle frontal | R |

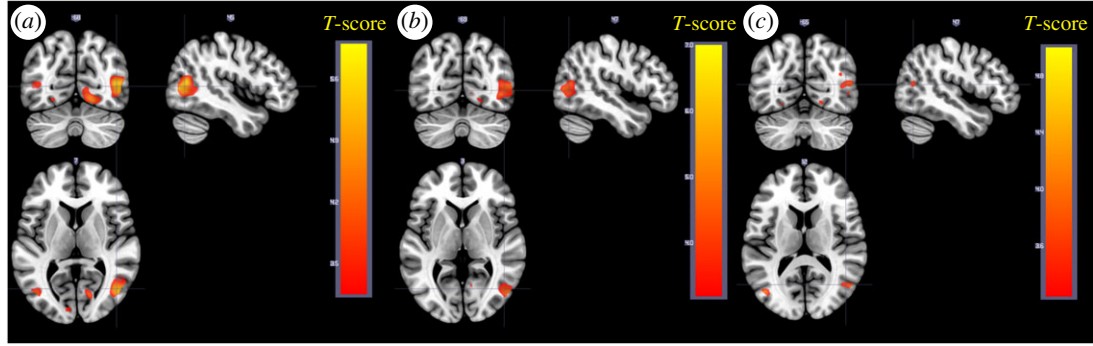

**Figure 5.** Upregulate > maintain. (*a*) All. (*b*) Young. (*c*) Older.

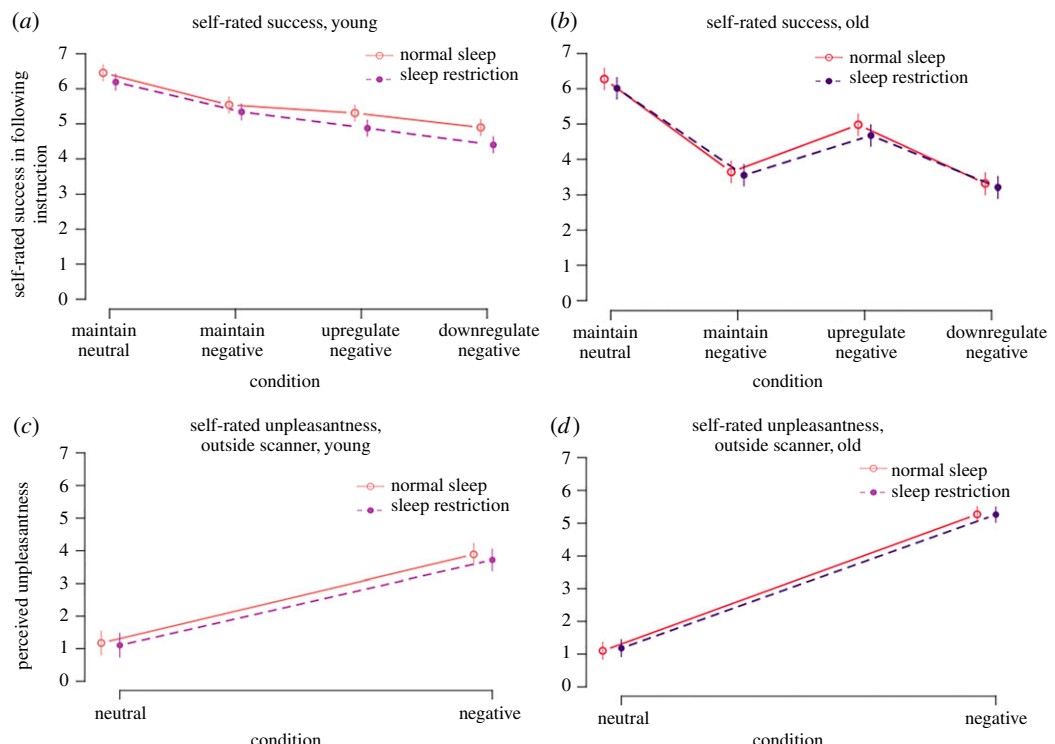

**Figure 6.** (*a*) Rated success in following the instructions. Younger and older participants behaved quite differently in the scanner. In young participants, sleep restriction caused a significant decrease in rated success for maintain neutral, downregulate and upregulate negative. Dots represent means and vertical lines 95% CI. (*b*) Rated unpleasantness. In younger participants, sleep restriction caused a borderline significant effect on ratings of unpleasantness in a subsample.

amygdala; left (0.14 [−0.05, 0.33], $p = 0.14$), right (0.14 [−0.05, 0.33], $p = 0.14$), right (0.08 [−0.05, 0.21], $p = 0.21$) or lOFC; left (0.11 [−0.17, 0.39], $p = 0.45$), right (0.07 [−0.12, 0.26], $p = 0.46$). The effect of sleep restriction on dlPFC was likewise not significant; left (0.07 [−0.09, 0.23], $p = 0.36$), right (0.06 [−0.16, 0.29], $p = 0.57$). Thus, the hypothesis that sleep restriction would be associated with decreased activation of dlPFC and lOFC and increased amygdala activation was not confirmed.

### 3.3.2. Effect of sleep restriction on upregulate > maintain negative (in young)

The effect of sleep restriction on the contrast upregulate > maintain was not significant in any cluster across the brain.

### 3.4. Summary of age effects

Full analyses are displayed in electronic supplementary material, figures S4–S12 and tables S2–S10. In sum, the main effect of age group on rated success was not significant (−0.18 [−0.51, 0.14], $p = 0.266$]

across stimulus types. However, age group and stimulus type interacted significantly ($p < 0.001$) such that older participants reported decreased success for maintain negative ($-1.77$ [$-1.27$, $-2.25$], $t_{67} = 7.16$, $p < 0.001$) and downregulate negative ($-1.29$ [$-0.84$, $-1.74$], $t_{72} = 5.68$, $p < 0.001$) compared to young, whereas there were no age differences for maintain neutral ($-0.18$ [$0.13$, $-0.49$], $t_{62} = 1.14$, $p = 0.257$) and upregulate negative ($-0.19$ [$0.26$, $-0.65$], $t_{54} = 0.85$, $p = 0.397$). Sleep restriction had the main effect on rated success in older in the direction of lower success after sleep restriction, but no interaction with stimulus type (see electronic supplementary material).

Age group had a big effect on ratings of unpleasantness (outside the scanner), with older participants reporting higher unpleasantness compared to young ($1.56$ [$1.16$, $1.96$], $p < 0.001$). Age group also interacted with valence ($p < 0.001$), in that older participants reported increased unpleasantness compared to young to negative stimuli ($1.25$ [$0.65$, $1.86$], $t_{25} = -4.25$, $p < 0.001$) but no difference was observed for neutral stimuli ($-0.07$ [$-0.19$, $0.05$], $t_{53} = -1.21$, $p = 0.23$). No effects of sleep restriction were significant (see electronic supplementary material).

For the contrast negative > neutral, young participants showed more activity in the occipital region compared to older (see electronic supplementary material, figures S4 and S5 and tables S2 and S3 for complete results). For the contrast downregulate > maintain, younger participants showed more activity around the frontal and precentral gyrus and also around the orbital part of superior frontal gyrus (see electronic supplementary material, figures S6 and S7 and tables S4 and S5). For upregulate > maintain, older participants showed more activity around the medial and superior temporal gyrus and in the paracentral lobule (electronic supplementary material, tables S6 and S7 and figures S8 and S9). In areas of interest, no voxels showed an effect of the age × sleep interaction for any of the contrasts.

Thus, in general, older participants showed less success for maintain compared to young and a brain activation pattern with less activity for downregulate and more for upregulate compared to young. Sleep restriction caused a general decrease in rated success, but no effects on brain activity.

## 3.5. Connectivity

To test the specific pre-registered hypothesis that sleep restriction would cause decreased connectivity between amygdala and dlPFC/lOFC, an ROI analysis was performed in dlPFC and lOFC (bilaterally) for the contrasts negative > neutral and downregulate > maintain (negative), for bilateral amygdalae in young participants (all participants presented in electronic supplementary material). In young participants, sleep restriction was not associated with any significant effect on any connectivity from amygdala to lOFC or dlPFC (see electronic supplementary material, table S11). At whole brain level, sleep restriction did not have any significant effect on the connectivity from amygdala to anywhere in the brain, either for negative > neutral or for downregulate > maintain, in young participants.

To study connectivity related to negative valence, a PPI analysis was performed for the contrast negative > neutral (maintain instruction) with seeds in bilateral amygdalae across all participants and sleep conditions. Negative, compared to neutral stimuli, caused an increase in connectivity between amygdala and occipital areas (fusiform and extrastriate) for both left and right amygdala (table 7 and figure 7).

To study the effect of downregulating on amygdala connectivity, a second PPI analysis was performed for the contrast downregulate > maintain (negative) with seeds in bilateral amygdala. Some small clusters of voxels showed an effect at $p = 0.001$ uncorrected, but none of them survived whole-brain correction and were therefore judged as random findings. All maps can be found at Neurovault (https://neurovault.org/collections/FWHMMCKI/).

## 4. Discussion

This study investigated the effects of sleep restriction on emotional regulation through cognitive reappraisal in older and younger participants. Sleep restriction caused younger participants to rate lower success in regulating their emotional response, and a tendency to perceive both neutral and negative stimuli as less unpleasant, but no effect was seen on neural correlates, i.e. amygdala activity or connectivity. Irrespective of sleep condition, young participants showed increased activity in dlPFC as well as in lOFC when downregulating, as expected, while this effect was not significant in older participants. However, older participants also displayed difficulties following the task instructions. Passive viewing of negative pictures, irrespective of sleep condition and age group, was associated

**Table 7.** PPI analysis. Negative > neutral.

| cluster equivk | peak p(FWE-corr) | peak T | peak equivZ | peak p(unc) | x | y | z (mm) | MRIcron AAL (peak) | side |
|---|---|---|---|---|---|---|---|---|---|
| A. Negative > neutral, left amygdala | | | | | | | | | |
| 4556 | 0.009 | 5.2 | 4.93 | <0.001 | 24 | −60 | −14 | fusiform | R |
| | 0.021 | 4.98 | 4.73 | <0.001 | −40 | −68 | −2 | middle occipital | L |
| | 0.152 | 4.4 | 4.23 | <0.001 | −26 | −63 | −16 | cerebellum | L |
| 497 | 0.259 | 4.22 | 4.06 | <0.001 | 38 | −86 | 12 | middle occipital | R |
| | 0.988 | 3.25 | 3.18 | 0.001 | 26 | −78 | 15 | n.a. | |
| 45 | 0.716 | 3.74 | 3.63 | <0.001 | −36 | −44 | −20 | fusiform | L |
| 68 | 0.758 | 3.7 | 3.59 | <0.001 | −30 | −92 | −6 | inferior occipital | L |
| B. Negative > neutral, right amygdala | | | | | | | | | |
| 671 | 0.098 | 4.53 | 4.34 | <0.001 | 45 | −66 | −6 | inferior temporal | R |
| | 0.673 | 3.76 | 3.65 | <0.001 | 45 | −57 | −14 | inferior temporal | R |
| | 0.986 | 3.25 | 3.18 | 0.001 | 52 | −64 | 3 | middle temporal | R |
| 493 | 0.285 | 4.17 | 4.02 | <0.001 | −46 | −63 | −2 | middle temporal | L |
| | 0.741 | 3.7 | 3.59 | <0.001 | −39 | −78 | −8 | inferior occipital | L |
| 451 | 0.368 | 4.07 | 3.93 | <0.001 | 34 | −81 | 18 | middle occipital | R |
| 35 | 0.501 | 3.93 | 3.8 | <0.001 | 38 | −16 | −9 | n.a. | |
| 64 | 0.736 | 3.7 | 3.59 | <0.001 | 22 | −24 | 2 | thalamus | R |
| 55 | 0.903 | 3.49 | 3.4 | <0.001 | −33 | −45 | −20 | fusiform | L |

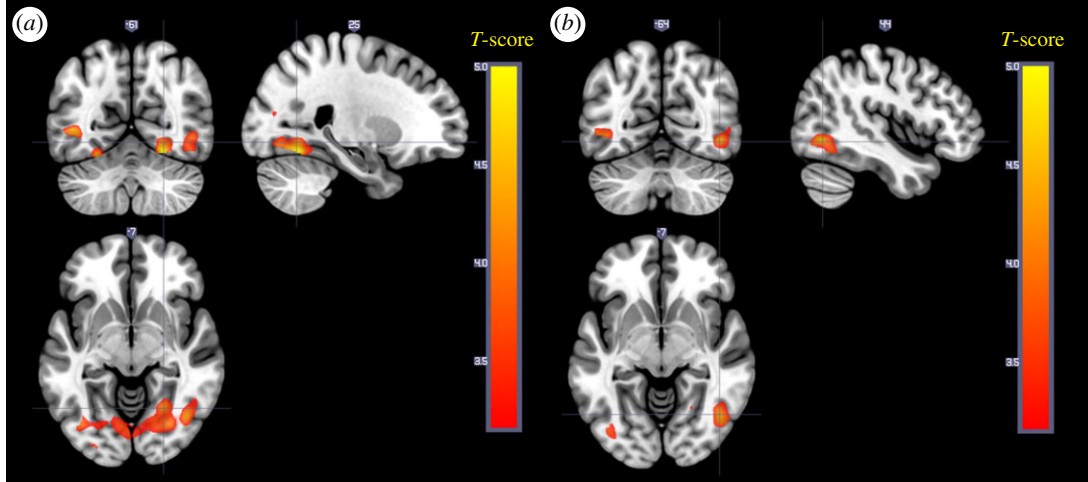

**Figure 7.** (*a*) Connectivity from left amygdala that increases for negative compared to neutral pictures. (*b*) Connectivity from right amygdala that increases for negative compared to neutral pictures.

with increased amygdala activity and increased connectivity to occipital and extrastriate cortex. Even though no measurable neural correlates were observed, sleep restriction was followed by impaired emotional regulation, further strengthening the notion that sleep is important for emotional reactivity and the degrees of control over affective responses and individual experiences.

Consistent with previous studies of cognitive reappraisal [11,16–18,20,36], an increased activation in lOFC (extending in to vlPFC) and dlPFC when downregulating was observed. This effect was, however, only significant in the younger group, who also to a higher degree indicated that they had followed the instructions. Amygdala responses to negative compared to neutral pictures could be shown when the stimuli were modelled with a short duration. This is coherent with the view that amygdala responses are primarily related to the onset of the stimulus. Indirect effects of amygdala activation, i.e. the enhancement of perception of emotional stimuli [37,38], are usually more apparent. In response to negative stimuli, we could indeed see an increased connectivity to visual cortical areas (fusiform, inferior and middle occipital), which fits with the idea of enhanced perception during negative affect. No increase in connectivity between amygdala and dlPFC and lOFC was seen for downregulating in any group, contrary to what was expected based on the findings from [20] and also no effect of downregulating was shown on amygdala. Possibly, this indicates that lOFC and dlPFC are not directly inhibiting the amygdala in cognitive reappraisal but are part of a more complex network. Upregulation was associated with increased activity in cingulate cortex, frontal areas and supplemental motor area, in line with the meta-analysis by Frank *et al.* [18]. Altogether, the task effects are consistent with previous studies in young. This was not the case for older participants, and as mentioned above, this was the reason why the effects of sleep restriction were primarily studied in young.

The main aim of the study was to investigate the effects of sleep restriction on emotion regulation. It has previously been proposed that sleep deprivation causes increased amygdala activation in response to negative stimuli [6,39] and that the mechanism behind this phenomenon is a prefrontal-amygdala disconnect [6,40,41]. We found no effect of sleep restriction on amygdala activity nor connectivity to negative stimuli for passive viewing. Furthermore, when explicitly instructing young participants to regulate their response, there was no effect of sleep restriction on brain activity or connectivity. When including the older adults in the analysis, there was even an increase in connectivity between amygdala and dlPFC and lOFC following sleep restriction (see electronic supplementary material). These findings were in contrast to the hypotheses based on the findings by Yoo *et al.* [6]. It should be noted that the well-cited study by Yoo and colleagues used a slightly different passive viewing task, with increasingly aversive stimuli and most importantly total sleep deprivation, with potentially stronger effects. However, the sample size in that study was smaller, and a between-group design was used, increasing the error variance and risk for confounding. To our knowledge, the number of studies showing similar amygdala effects is so far limited [7,39,41] and no study appears to have replicated the findings with a similar design. One possible explanation for the lack of amygdala change after sleep restriction in the present study is that the partial sleep restriction procedure,

compared to previous studies, allowed participants to have enough REM sleep, occurring mainly at the end of the night, for the emotional processing it is believed to subserve [42]. Another possible cause may be that some of the subjects were partially sleep deprived in the full sleep condition and that the difference between the conditions was not enough to cause changes in the brain activity or connectivity. The finding that sleep restriction was associated with lower self-reported regulation success underlines the importance of sleep for emotional functioning, and further efforts to understand brain correlates to this association are called for.

As previously mentioned, some data indicate an association between long-term poor sleep quality (or use of sleep medication) and lower ability in a reappraisal task [22,23]. A meta-analysis also showed that patients with several psychiatric disorders that include sleep disturbances show less brain activity and to some extent decreased self-reported success in cognitive reappraisal [43]. Interestingly, in the, to our knowledge, largest study of sleep quality and amygdala reactivity, a positive association between bilateral amygdala reactivity and measures of depressive symptoms and perceived psychological stress was found in participants reporting poor overall sleep, but not in good sleepers [44]. A possible interpretation of these findings is that a longer period (than one night) of disturbed sleep is needed to cause potential morphological or functional changes in the underlying brain structures involved in emotional regulation. It is also possible that sample differences in sensitivity to sleep restriction explain the differences between our sample and previous studies [6,7]. One such difference that we aimed to address in this study was age. Some of our previous work indicates that the effect of sleep restriction on both empathy [33] and mood [45] is different in older age. This study did not specifically analyse the interaction between sleep restriction and age, but the results on ratings of success and unpleasantness are in line with a potentially reduced sensitivity to sleep loss in older.

After the session, the participants were asked what strategy they used to regulate their emotion. The main purpose of this was to evaluate whether the participants were able to follow the instructions. We excluded participants who obviously did not follow the instructions, but for a larger sample of the older group, we could not exclusively judge whether this was the case since they were unable to precisely specify what strategy they used. Younger and older participants were also indicated to be differently successful in performing the cognitive reappraisal task, according to ratings of success. The results for the older age group should therefore be interpreted with caution, and for this reason, we focused the analyses on the younger participants. Nonetheless, older participants reported higher unpleasantness to negative, but not neutral stimuli, compared to young. Older participants also rated lower success in maintaining (passive viewing) negative compared to neutral stimuli. A possible interpretation is that older participants generally have a bias for positive stimuli in attention and memory, known as the positivity effect in older [46], and therefore had a hard time passively viewing the negative stimuli without controlling the response. When contrasting downregulating to maintain negative stimuli in older participants only, the expected responses in dlPFC and lOFC were not significant. This could also be caused by the fact that older participants spontaneously regulate their emotion in response to the negative stimuli resulting in a less effective contrast, and hence there is no difference when explicitly asked to regulate.

## 4.1. Strength and limitations

Statistical power is a general issue of consideration in neuroimaging studies [47,48]. Here, a within-subjects design was used to reduce error variance, and the sample size was larger than in previous experimental studies of sleep and amygdala reactivity [6,39]. Still, power may have been too low to detect effects of interest, especially with the use of partial compared to total sleep deprivation. Age effects were hard to determine, and importantly, putative effects from the cross-sectional, non-random samples could be due to generation effects rather than effects of age *per se*. It could also be argued that ratings of subjective unpleasantness or similar would have been a more relevant behavioural outcome than ratings of success. It should also be noted that the physiological outcomes (heart rate and pupil diameter) did not support the behavioural findings of decreased emotional regulation success. Regarding the stimuli, the IAPS pictures were not balanced/controlled for luminance, possibly contributing to error variance in the effects of stimuli on pupil diameter as well as fMRI effects in the visual cortex. Even though this is the first study combining subjective ratings and brain imaging measuring in an emotional regulation task investigating the effect of restricted sleep, methodological developments are called for, hopefully also involving future studies across age groups to improve generalizability.

# 5. Conclusion

In conclusion, the present study corroborates the importance of sleep for emotional regulation by showing that an ecologically relevant model of suboptimal sleep—when restricted to three hours of sleep—still negatively affects the capacity for emotional regulation. The negative effect of sleep restriction on self-rated emotional regulation success was, however, not paralleled by any significant effects in amygdala activity or connectivity, potentially calling into question the idea of a prefrontal-amygdala disconnect as a mechanism for the effect of sleep deprivation on emotional regulation. Further understanding of neural mechanisms underlying the behavioural findings might help to clarify the role of suboptimal sleep in conditions and disorders that are characterized by insufficient capacity for emotional regulation.

Ethics. The study was approved by the Regional Ethics Review board of Stockholm (2012/1870-32) and preregistered at clinicaltrials.gov (NCT02000076) with a separate hypotheses list published at Open Science Framework (https://osf.io/bxfsb/). All participants gave written informed consent.

Data accessibility. Structural and functional imaging data and pupil diameter and heart rate are available at https://openfmri.org/dataset/ds000201/. Code for stimulus presentation, preprocessing and analysis, and ratings are found at: https://doi.org/10.5281/zenodo.235595 and https://doi.org/10.5281/zenodo.1434679. https://doi.org/10.5281/zenodo.235595 and https://doi.org/10.5281/zenodo.1434679. Statistical maps can be found at https://neurovault.org/collections/FWHMMCKI/.

Authors' contributions. G.N., S.T., J.S., H.F., G.K., M.L., A.G., T.Å. designed the study. S.T, G.N. acquired the data. S.T. analysed the data. S.T., G.N., M.L., J.S., H.F., G.K., A.G., P.P., T.Å. interpreted the results. S.T. drafted the manuscript. All authors read and approved the final version of the manuscript.

Competing interests. The authors declare no competing interests

Funding. This work was funded by Stockholm Stress Center, Riksbankens Jubileumsfond, Karolinska Institutet, Stockholm County Council, Isabella and Henrik Berg and the Heumanska stiftelsen/Hjärnfonden, Fredrik and Ingrid Thuring's Foundation.

Acknowledgements. We are thankful to Hanna Thuné, Paolo d'Onofrio, Diana Sanchez Cortes, Danielle Cosme, Birgitta Mannerstedt Fogelfors and Roberta Nagai for assistance with the data collection. We are also thankful to Jonathan Berrebi and Rouslan Sitnikov for technical advice.

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
