## [Reviewer comments · Royal Society Open Science]

Review History

RSOS-181704.R0 (Original submission)

Review form: Reviewer 1

Is the manuscript scientifically sound in its present form?

Yes

Are the interpretations and conclusions justified by the results?

No

Is the language acceptable?

Yes

Is it clear how to access all supporting data?

Yes

Do you have any ethical concerns with this paper?

No

Have you any concerns about statistical analyses in this paper?

No

Recommendation?

Accept with minor revision (please list in comments)

Comments to the Author(s)

This report is one of many arising from the Stockholm Sleepy Brain Project. The strengths of the program are that they have a good number of participants performing standardized tasks designed to test specific cognitive or affective constructs. The authors pre-register their expectations of the data. Data are processed with considerable care and finesse. The reporting of results is of high quality.

In this installment, the authors examined emotional regulation of emotional face pictures and either kept a neutral stance, enhanced or suppressed their emotional response to a given picture. Young and old adults were studied. The condition of interest was restriction of sleep to 3h/night for a single night. The authors collected subjective self-reports of ability to regulate emotion as well as imaging data which examined both activation magnitude and functional connectivity assessed using PPI methodology. Specific ROIs were targeted during analysis.

The documentation of the motivation for the work and the description of the experiment and the result presentation were admirably clear. Although the study was a negative one especially with respect to the age / sleep loss interaction, I believe that its strengths alluded to earlier merit eventual publication of the findings.

Major comments:

1. The task may be one that is widely used, but I'm concerned by the absence of objective behavioral and physiologic findings to back up their assertion of impaired regulation of emotional response to pictures with sleep restriction. The behavioral limitation arises from a task that only collects subjective ratings. The lack of physiological support was not for want of trying. The authors collected HR and pupil diameter measures, but the error variance precluded finding effects worthy of reporting in the main text. This should prompt the authors to soften their stance on their interpretation of the neuroimaging findings and in casting the title. It seems as if they are casting doubt on the validity of previous work by Yoo (2007), Motomura (2013) and Prather (2013) in the title and discussion.
2. The imaging results showing the effects of the negative stimulus and the effects of regulation but do not show the hypothesized effects of sleep restriction. I believe that contributing to this finding is that previous work was based on stronger manipulation of sleep: a single night of total sleep deprivation (Yoo 2007) or 5 nights of restriction to 4h/night TIB (Motomura) or multiple nights of habitual (poorer) sleep quality (Prather). This point is related to #1 - a failure to discover an effect found by others should not be taken as the mechanism proposed by earlier work to be invalid. This rather than the low power (this is not the case!) of the study is likely to be why this and a previous imaging study showed negative findings relative to prior literature. At the very least, this possibility should be added to the list of limitations.

Review form: Reviewer 2

Is the manuscript scientifically sound in its present form?

Yes

Are the interpretations and conclusions justified by the results?

Yes

Is the language acceptable?

Yes

Is it clear how to access all supporting data?

Yes

Do you have any ethical concerns with this paper?

No

Have you any concerns about statistical analyses in this paper?

No

Recommendation?

Accept with minor revision (please list in comments)

Comments to the Author(s)

Tamm and colleagues have investigated the effects of sleep restriction on emotional regulation in old and young participants. They found hemodynamic changes and behavioral results somewhat consistent with the previous literature, yet no effect of sleep restriction on fMRI-measured activations.

The article is presented in a concise but detailed and honest manner. I believe the result is interesting for researchers in the field. My main comment concerns the misunderstanding of the instructions and the possibility that this appeared reflected in the fMRI results. I'd like to ask the authors to explain with more detail the nature of the misunderstanding by the older participants. This has the benefit of warning future researchers of potential pitfalls that might have gone unnoticed so far, especially considering that the rate of misunderstanding of the instructions seems to be higher than in other previous publications. Also, I wonder if the self-assessed rating of the success by the participants could be used to separate high vs. low success trials and run fMRI covariates with each separate division. It would be interesting to see how the success relates to the fMRI activations and whether sleep-related changes are contingent to one or the other group.

Decision letter (RSOS-181704.R0)

12-Feb-2019

Dear Mrs Tamm

On behalf of the Editors, I am pleased to inform you that your Manuscript RSOS-181704 entitled "Sleep restriction caused impaired emotional regulation without detectable brain activation changes - a functional magnetic resonance imaging study" has been accepted for publication in

Royal Society Open Science subject to minor revision in accordance with the referee suggestions. Please find the referees' comments at the end of this email.

The reviewers and handling editors have recommended publication, but also suggest some minor revisions to your manuscript. Therefore, I invite you to respond to the comments and revise your manuscript.

- Ethics statement

- Data accessibility

If you wish to submit your supporting data or code to Dryad (<http://datadryad.org/>), or modify your current submission to dryad, please use the following link:
<http://datadryad.org/submit?journalID=RSOS&manu=RSOS-181704>

- Competing interests

- Authors' contributions

- Acknowledgements

- Funding statement

Because the schedule for publication is very tight, it is a condition of publication that you submit the revised version of your manuscript before 21-Feb-2019. Please note that the revision deadline will expire at 00.00am on this date. If you do not think you will be able to meet this date please let me know immediately.

Supplementary files will be published alongside the paper on the journal website and posted on the online figshare repository (<https://rs.figshare.com/>). The heading and legend provided for each supplementary file during the submission process will be used to create the figshare page,

so please ensure these are accurate and informative so that your files can be found in searches. Files on figshare will be made available approximately one week before the accompanying article so that the supplementary material can be attributed a unique DOI.

on behalf of Dr Anastasia Christakou (Associate Editor) and Antonia Hamilton (Subject Editor)
openscience@royalsociety.org

Reviewer comments to Author:
Reviewer: 1

Comments to the Author(s)

This report is one of many arising from the Stockholm Sleepy Brain Project. The strengths of the program are that they have a good number of participants performing standardized tasks designed to test specific cognitive or affective constructs. The authors pre-register their expectations of the data. Data are processed with considerable care and finesse. The reporting of results is of high quality.

In this installment, the authors examined emotional regulation of emotional face pictures and either kept a neutral stance, enhanced or suppressed their emotional response to a given picture. Young and old adults were studied. The condition of interest was restriction of sleep to 3h/night for a single night. The authors collected subjective self-reports of ability to regulate emotion as well as imaging data which examined both activation magnitude and functional connectivity assessed using PPI methodology. Specific ROIs were targeted during analysis.

The documentation of the motivation for the work and the description of the experiment and the result presentation were admirably clear. Although the study was a negative one especially with respect to the age / sleep loss interaction, I believe that its strengths alluded to earlier merit eventual publication of the findings.

Major comments:

1. The task may be one that is widely used, but I'm concerned by the absence of objective behavioral and physiologic findings to back up their assertion of impaired regulation of emotional response to pictures with sleep restriction. The behavioral limitation arises from a task that only collects subjective ratings. The lack of physiological support was not for want of trying. The authors collected HR and pupil diameter measures, but the error variance precluded finding effects worthy of reporting in the main text. This should prompt the authors to soften their stance on their interpretation of the neuroimaging findings and in casting the title. It seems as if they are casting doubt on the validity of previous work by Yoo (2007), Motomura (2013) and Prather (2013) in the title and discussion.

2. The imaging results showing the effects of the negative stimulus and the effects of regulation but do not show the hypothesized effects of sleep restriction. I believe that contributing to this finding is that previous work was based on stronger manipulation of sleep: a single night of total sleep deprivation (Yoo 2007) or 5 nights of restriction to 4h/night TIB (Motomura) or multiple nights of habitual (poorer) sleep quality (Prather). This point is related to #1 - a failure to discover an effect found by others should not be taken as the mechanism proposed by earlier work to be invalid. This rather than the low power (this is not the case!) of the study is likely to be why this and a previous imaging study showed negative findings relative to prior literature. At the very least, this possibility should be added to the list of limitations.

Reviewer: 2

Comments to the Author(s)

Tamm and colleagues have investigated the effects of sleep restriction on emotional regulation in old and young participants. They found hemodynamic changes and behavioral results somewhat consistent with the previous literature, yet no effect of sleep restriction on fMRI-measured activations.

The article is presented in a concise but detailed and honest manner. I believe the result is interesting for researchers in the field. My main comment concerns the misunderstanding of the instructions and the possibility that this appeared reflected in the fMRI results. I'd like to ask the authors to explain with more detail the nature of the misunderstanding by the older participants. This has the benefit of warning future researchers of potential pitfalls that might have gone unnoticed so far, especially considering that the rate of misunderstanding of the instructions seems to be higher than in other previous publications. Also, I wonder if the self-assessed rating of the success by the participants could be used to separate high vs. low success trials and run fMRI covariates with each separate division. It would be interesting to see how the success relates to the fMRI activations and whether sleep-related changes are contingent to one or the other group.

Author's Response to Decision Letter for (RSOS-181704.R0)

See Appendix A.

Decision letter (RSOS-181704.R1)

21-Feb-2019

Dear Mrs Tamm,

I am pleased to inform you that your manuscript entitled "Sleep restriction caused impaired emotional regulation without detectable brain activation changes – a functional magnetic resonance imaging study" is now accepted for publication in Royal Society Open Science.

on behalf of Dr Anastasia Christakou (Associate Editor) and Professor Antonia Hamilton (Subject Editor)
openscience@royalsociety.org

Appendix A

We would like to thank the reviewers for their thorough review of our manuscript and for the constructive comments and suggestions, which have helped improve the manuscript. Our responses are stated below in *italics*. All changes are marked with yellow in the manuscript.

Reviewer 1.

1. The task may be one that is widely used, but I'm concerned by the absence of objective behavioral and physiologic findings to back up their assertion of impaired regulation of emotional response to pictures with sleep restriction. The behavioral limitation arises from a task that only collects subjective ratings. The lack of physiological support was not for want of trying. The authors collected HR and pupil diameter measures, but the error variance precluded finding effects worthy of reporting in the main text. This should prompt the authors to soften their stance on their interpretation of the neuroimaging findings and in casting the title. It seems as if they are casting doubt on the validity of previous work by Yoo (2007), Motomura (2013) and Prather (2013) in the title and discussion.

We agree with the reviewer that the lack of effects on other behavioral and psychologic measures apart from self-rated success is indeed a shortcoming of the present study. As suggested, we have now slightly changed the wording in the discussion (lines 502-503, 508, 557, 561-563) and the conclusion (line 576).

2. The imaging results showing the effects of the negative stimulus and the effects of regulation but do not show the hypothesized effects of sleep restriction. I believe that contributing to this finding is that previous work was based on stronger manipulation of sleep: a single night of total sleep deprivation (Yoo 2007) or 5 nights of restriction to 4h/night TIB (Motomura) or multiple nights of habitual (poorer) sleep quality (Prather). This point is related to #1 – a failure to discover an effect found by others should not be taken as the mechanism proposed by earlier work to be invalid. This rather than the low power (this is not the case!) of the study is likely to be why this and a previous imaging study showed negative findings relative to prior literature. At the very least, this possibility should be added to the list of limitations.

We agree with the reviewer that the difference in the type of sleep manipulation could indeed been an explanation behind the differences between this and previous studies. This is now specifically commented on in the discussion (lines 500-501) and limitations (line 555). We have also made it clear in the abstract that this study used partial sleep restriction.

Reviewer: 2

Comments to the Author(s)

Tamm and colleagues have investigated the effects of sleep restriction on emotional regulation in old and young participants. They found hemodynamic changes and behavioral results somewhat consistent with the previous literature, yet no effect of sleep restriction on fMRI-measured activations.

The article is presented in a concise but detailed and honest manner. I believe the result is interesting for researchers in the field. My main comment concerns the misunderstanding of the instructions and the possibility that this appeared reflected in the fMRI results. I'd like to ask the authors to explain with more detail the nature of the misunderstanding by the older participants. This has the benefit of warning future researchers of potential pitfalls that might have gone unnoticed so far, especially considering that the rate of misunderstanding of the instructions seems to be higher than in other previous publications.

This is indeed an important point and we have added a short explanation in the method section (lines 165-167).

Also, I wonder if the self-assessed rating of the success by the participants could be used to separate high vs. low success trials and run fMRI covariates with each separate division. It would be interesting to see how the success relates to the fMRI activations and whether sleep-related changes are contingent to one or the other group.

We agree with the reviewer that it would be interesting to relate the self-rated success to fMRI results. However, intra-individual variability for each type of instruction was low, limiting the scope for analyses of this kind.